# LoCoHD: a metric for comparing local environments of proteins

Zsolt Fazekas [1,2], Dóra K. Menyhárd [1,3] & András Perczel [1,3] ✉

Protein folds and the local environments they create can be compared using a variety of differently designed measures, such as the root mean squared deviation, the global distance test, the template modeling score or the local distance difference test. Although these measures have proven to be useful for a variety of tasks, each fails to fully incorporate the valuable chemical information inherent to atoms and residues, and considers these only partially and indirectly. Here, we develop the highly flexible local composition Hellinger distance (LoCoHD) metric, which is based on the chemical composition of local residue environments. Using LoCoHD, we analyze the chemical heterogeneity of amino acid environments and identify valines having the most conserved-, and arginines having the most variable chemical environments. We use LoCoHD to investigate structural ensembles, to evaluate critical assessment of structure prediction (CASP) competitors, to compare the results with the local distance difference test (lDDT) scoring system, and to evaluate a molecular dynamics simulation. We show that LoCoHD measurements provide unique information about protein structures that is distinct from, for example, those derived using the alignment-based RMSD metric, or the similarly distance matrix-based but alignment-free lDDT metric.

The Research Collaboratory of Structural Bioinformatics Protein Data Bank (RCSB-PDB)[1] currently contains more than 217,000 experimentally determined protein structures. As the function of a protein is closely linked to its structure, this database provides valuable information on biological processes related to evolution, development, disease progression, drug design or agriculture, to name a few. In order to understand the behavior of these 3D atomic arrangements, computational tools and algorithms have been developed for their numerical analysis. In silico methods, such as molecular dynamics protocols[2–4], molecular modeling software[5–7], AI-based systems[8–10], and de novo protein design platforms[11,12] have also become increasingly popular and accessible, generating a vast amount of structural data concerning bio-macromolecular systems.

The comparison of different conformations of the same protein, or of the structures of similarly folded but different proteins, can be realized either by metrics, by generalized metrics (which are less

restricted), or by similarity measures (for an overview of some of the available methods[13–31], please refer to Supplementary Note 1 and Supplementary Table 1). However, if the intention is to compare two proteins based on the chemical environment of their components, the commonly used measures do not provide focused information. The appearance or disappearance of different side-chain interactions (or interaction networks), changes in salt bridges, hydrogen bonds, π-cation interactions, polar-polar contacts, hydrophobic cores are all vital information. Since the nature of these environments dictates how proteins fold, move, or interact with each other, it should be critical to develop a method that provides an objective measure for their characterization.

With this in mind, we developed the local composition Hellinger distance (LoCoHD) metric presented here, which measures the chemical and structural difference between two local environments in proteins. We aim to provide a highly flexible scheme for objective

[1]Laboratory of Structural Chemistry and Biology, Institute of Chemistry, ELTE Eötvös Loránd University, Budapest, Hungary. [2]ELTE Hevesy György PhD School of Chemistry, ELTE Eötvös Loránd University, Budapest, Hungary. [3]HUN-REN-ELTE Protein Modeling Research Group, ELTE Eötvös Loránd University, Budapest, Hungary. ✉e-mail: perczel.andras@ttk.elte.hu

comparison of two arbitrary atomic arrangements within a protein, while keeping the evaluation simple, intuitive, and relatively fast.

## Results

### Distribution of LoCoHD scores

In order to assign meaning to the absolute size of a score, and to decide whether a given score is "large" or "small", it is important to know the underlying distribution from which the score is sampled. Therefore, we set out to determine how the LoCoHD scores for the FA+Cent and CG+Cent typing schemes are distributed when the uniform weight function is used between 3 Å and 10 Å (see the Methods section, Description of the LoCoHD Algorithm subsection for clarification). We chose these schemes for this initial investigation because the residue centroid primitive atoms can serve as anchors when comparing any residue type with any other. This feature is necessary for the random sampling protocol described in the Methods section (Random LoCoHD distribution generation subsection). The distributions of these random samples are shown in Supplementary Fig. 1.

Although theoretically LoCoHD scores can range from 0 to 1, it can be seen that even random residue-pairs do not frequently achieve values greater than 40%. The resulting experimental distributions can be modeled with β-distributions with parameters $\alpha = 10.52$ and $\beta = 33.48$ (p-value is $2.22*10^{-12}$ according to the Kolmogorov-Smirnov test) for typing scheme FA+Cent, and parameters $\alpha = 12.99$ and $\beta = 35.48$ (p-value = $1.70*10^{-9}$) for typing scheme CG+Cent. For FA+Cent, the mean LoCoHD value is around 23.98% with a standard deviation (StDev) of 6.37%, and data ranging from 5.38% to 79.23%. For CG+Cent on the other hand, the mean LoCoHD value is around 26.83% with a StDev of 6.27%, and data ranging from 5.25% to 62.28%. It is important to keep in mind, that these distributions come from sampling random residue-pairs, which means that usually not the same amino acid types are paired and compared, likely resulting in a higher average LoCoHD score. The residue-pairs showing the lowest and highest LoCoHD scores in case of the FA+Cent typing were also extracted from the process. The lowest value (5.38%) belongs to the residue pair of PDB ID 2IJX (a PCNA3 monomer[32]), chain C, residue Ala[17] and PDB ID 1XHK (an ATP-dependent Lon protease[33]), chain B, residue Ala[501]. Both of these residue environments are hydrophobic cores, containing mostly aliphatic side-chains from valines, leucines, and isoleucines. The highest value (79.23%) belongs to the residue pair of PDB ID 6JV7 (a rat complement protein[34]), chain B, residue Gly[28] and PDB ID 3PL0 (a PF10014 dioxygenase[35]), chain A, residue Ile[143]. The isoleucine's environment (10 Å around the residue) in 3PL0 contains a significant amount of aromatic carbon primitive types coming from 10 different aromatic residues. This environment also has a relatively low charged primitive atom content, coming from only 4 residues. In contrast, the environment of glycine from 6JV7 contains 3 disulfide bridges and charged primitive types coming from 8 different residues. This way, the primitive type distribution of the two environments highly differ, resulting in the extremely high LoCoHD score.

Statistical descriptors for the different residue type pairs were also extracted from the random samples. The residue type pairs with the highest and lowest average LoCoHD scores are shown in Table 1 for the primitive typing scheme FA+Cent. A t-distributed stochastic neighborhood embedding (tSNE) was also performed using the mean LoCoHD of hetero-residue pairs (i.e. where the residue types are not the same), the result of which is shown in Fig. 1. Using this technique, we were able to map the 20 proteinogenic residues into a two-dimensional space, while preserving the topology dictated by their environmental similarities.

Our analysis shows that the LoCoHD score can distinguish between environments surrounding residues with different physicochemical properties and different environment-organizing behaviors. The residue pair Val-Thr is the first hetero-residue pair in Table 1, i.e. it has the lowest average LoCoHD score. This means that, on average, the

environments of Val and Thr are very similar both in composition and arrangement. This phenomenon is due to the isoelectronic and isosteric relationship between these two amino acids. The homo-residue pair with the highest average LoCoHD score is arginine, an indication of the diversity of its environments; the arginine side-chains can be solvated in the bulk solvent, can participate in H-bonds, salt-bridges, and π-cation interactions through their guanidino groups, and can participate in hydrophobic interactions through the Cβ-Cγ-Cδ aliphatic chain. Arginine also has the highest average LoCoHD scores calculated against all other amino acids. The amino acid with the most similar environments to arginine environments is lysine, followed by glutamine and tyrosine. The high environmental similarity between arginine and lysine is easily explained by their positive charge (see Fig. 1, box C). For further examples, see sections Supplementary Note 2 and Supplementary Figs. 2 and 3 of Supporting Info.

The tSNE analysis in Fig. 1 provides us a visual aid for noticing patterns in the 20 by 20 residue-residue average LoCoHD matrix. Points on this scatter plot represent individual amino acid types, while inter-point distances correlate with the average residue-residue environment dissimilarities. Besides the previously mentioned patterns, other, otherwise intuitive relationships can also be observed. Some residues stay close together, like the residue-sets Glu-Gln-Lys-Arg, Ile-Leu-Phe-Tyr-Trp, or Met-His. It is easy to find common physicochemical patterns in these close amino acids: Lys and Arg are both long, positively charged residues, Ile-Leu-Phe-Tyr-Trp are all hydrophobic residues, with Phe-Tyr-Trp forming a sub-cluster and having aromatic side-chains, and Met-His are common metal-complexing residues. Another noticeable pattern is formed by the Gly-Pro-Ala triplet, far away from all other amino acids. These are the residues with the smallest relative surface areas (with Ser included)[36] and are known to disrupt secondary structural elements.

To further validate the connection between the LoCoHD score of two residue environments and the presence/absence of secondary chemical interactions these residues participate in, we wanted to connect these properties through a machine learning model. It is evident that a good performing model can only be constructed, if there is a learnable connection between the inputs and the desired output. We trained a Siamese feedforward neural network that inputs the interaction fingerprints of two residues and tries to output the LoCoHD belonging to those residue environments. An interaction fingerprint vector contains the type of the residue (out of the 20 possibilities) and the number of different interactions it forms (6 counting-dimensions: H-bonds, van der Waals interactions, disulfide bonds, salt bridges, π-π stacking and π-cation interactions). It is important to note that this vector does not count the interactions that are present in the environment but are not formed by the central residue. Using these inputs the model was able to approximate the LoCoHD score (typing scheme: FA+Cent) of the two residue environments with a root mean squared error of 5.57% (validation: 5.57%) and a mean absolute error of 4.45% (validation: 4.39%). The final SpR between the predicted and true LoCoHD values came to be 0.454 (Supplementary Fig. 4). This result shows that a strong connection can be drawn between the LoCoHD score and the interaction fingerprint of the two central residues, but the information provided by the fingerprints is far from complete for an accurate score prediction.

Based on LoCoHD scores it was also possible to assess the environmental changes caused by functionally significant single mutations (and the absence of such in case of benign substitutions, see Supplementary Note 3-4 and Supplementary Figs. 5−8).

### Comparison of CASP14 contestants through LoCoHD and lDDT

We tested the performance of five CASP14 contestants against the LoCoHD scoring system and compared these results to lDDT scoring, one of the scores used in the competition. 31 target structures were collected from the CASP14 archive. We compared the environment of

**Table 1 | Rows show the residue-type pairs having the smallest 5 and largest 5 mean LoCoHD scores**

| Type pair | Number of samples | Mean | Median | StDev | Confidence interval | Minimum | Maximum |
|---|---|---|---|---|---|---|---|
| Val-Val | 2507 | 17.47% | 16.76% | 4.97% | 0.16% | 6.44% | 38.74% |
| Val-Thr | 3875 | 18.53% | 17.95% | 5.27% | 0.14% | 6.80% | 45.36% |
| Ala-Ala | 3105 | 18.60% | 17.92% | 5.45% | 0.16% | 5.38% | 41.35% |
| Ile-Ile | 1633 | 18.90% | 18.16% | 5.29% | 0.22% | 7.50% | 43.40% |
| Val-Ile | 3928 | 18.94% | 18.41% | 5.08% | 0.13% | 6.78% | 40.22% |
| ... 200 additional rows... | | | | | | | |
| Asp-Arg | 3074 | 29.31% | 28.98% | 6.05% | 0.18% | 13.48% | 52.51% |
| Gly-Arg | 3710 | 29.43% | 28.94% | 6.35% | 0.17% | 11.69% | 55.33% |
| Ser-Arg | 3281 | 29.56% | 29.30% | 6.37% | 0.18% | 13.39% | 56.21% |
| Pro-Arg | 2289 | 29.81% | 29.36% | 6.34% | 0.22% | 13.58% | 63.09% |
| Cys-Arg | 762 | 29.94% | 29.58% | 6.08% | 0.36% | 11.81% | 48.34% |

Primitive typing scheme FA+Cent with residue centroid anchors and a uniform 3 Å to 10 Å weight function were used. Two sided confidence intervals were calculated with a 95% upper confidence bound. Source data are provided as a Source Data file.

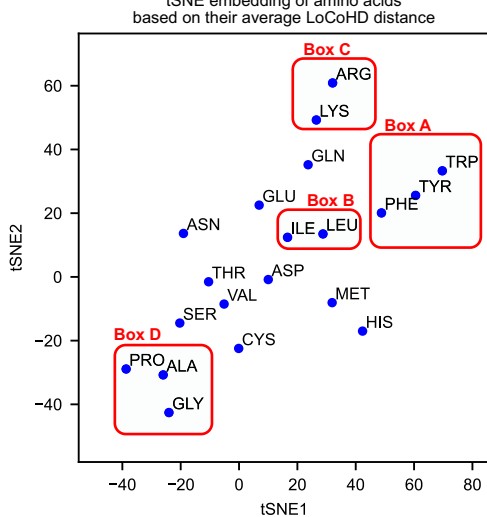

**Fig. 1 | Average LoCoHD-based residue embedding.** The t-distributed stochastic neighborhood embedding (tSNE) of the 20 proteinogenic residues based on the average LoCoHD scores of their environments for the primitive FA+Cent typing scheme. It can be seen that LoCoHD scores can cluster together environments around residues with similar physicochemical properties. In the embedding the large aromatic residues (Phe, Tyr, and Trp) form a cluster (box **A**), the highly similar Leu and Ile are close together (box **B**), the two positive residues Arg and Lys form a close pair (box **C**), and the small, secondary structure disrupting residues (Gly, Ala, Pro) also form a close triple (box **D**). Source data are provided as a Source Data file.

**Table 2 | lDDT and LoCoHD scoring statistics for the first five CASP14 contestants**

| | AlphaFold2 (TS427) | BAKER (TS473) | BAKER-experimental (TS403) | FEIG-R2 (TS480) | Zhang (TS129) |
|---|---|---|---|---|---|
| Median SpR(lDDT, LoCoHD) | −0.6788 (±0.1106) | −0.5550 (±0.1239) | −0.5257 (±0.1279) | −0.5071 (±0.1632) | −0.4847 (±0.1583) |
| Median prm-lDDT | 0.8410 (±0.0884) | 0.6000 (±0.1344) | 0.5860 (±0.1304) | 0.5350 (±0.1471) | 0.5210 (±0.1347) |
| Median prm-LoCoHD | 0.0814 (±0.0177) | 0.1311 (±0.0216) | 0.1340 (±0.0211) | 0.1425 (±0.0215) | 0.1482 (±0.0213) |

In this table the median Spearman's correlation coefficient (SpR), per-residue median (prm-) lDDT and prm-LoCoHD values are reported over all datasets, i.e. protein structures. The standard deviation of these values are presented between parentheses. It can be observed, that the lDDT and LoCoHD scoring systems agree on the order of the five contestants (rows median prm-lDDT and prm-LoCoHD). Also, it can be seen that as the quality of the prediction decreases (lDDT decreases, LoCoHD increases) the magnitude of the median SpR value decreases (row median SpR). Source data are provided as a Source Data file.

each true (experimental) and predicted residue for each contestant and target structure. This was done using the primitive FA+Cent typing scheme with a uniform weight function between 3 Å and 10 Å. Only hetero-residue contacts were allowed. The geometric center of each residue was used as the anchor atom.

We generated a dataset for each predicted structure, where each data point in the dataset belongs to one residue. The data points are two-dimensional vectors, with the residue's LoCoHD score as its first coordinate and the residue's lDDT score as its second coordinate. Different statistical descriptors were then generated for each dataset, namely the per-residue median of the LoCoHD and lDDT values (denoted prm-LoCoHD and prm-lDDT, respectively), and also their Spearman's rank correlation coefficient (SpR). The full statistics of these descriptors are reported in Supplementary Table 2. Here, we focus only on the median values, which can be found in Table 2. We

also collected all data points into two-dimensional histograms, one for each predictor. These are depicted in Fig. 2.

Our first results show an agreement between the contestant-order set by the median lDDT and LoCoHD values, proving that LoCoHD is able to separate (true, predicted) structure pairs with high, but different similarities. This is to be expected from a proper scoring system, as similarity scores should converge to their maximum values as the similarity between the structure pair increases, while dissimilarity scores should tend towards their minimum value. These tendencies inherently create correlations between different scoring systems, with higher absolute correlations closer to similarity extremities. This effect is clearly reflected in the median SpR values in Table 2. For the best-performer AlphaFold2, the median SpR value among the different structures is around −0.679, indicating a relatively high correlation. As we progress downwards on the list however, the SpR steadily decreases down to −0.485, supporting the aforementioned statement.

We also inspected some special cases individually. Since for every CASP14 predictor there are five predicted structures belonging to one target (experimental) structure, we can select target structures for which the predictions are outstandingly different. Namely, for every predictor, we identified the target structures for which the predicted structures show the largest SpR, median LoCoHD, and median lDDT gaps. These can be seen in Supplementary Table 3.

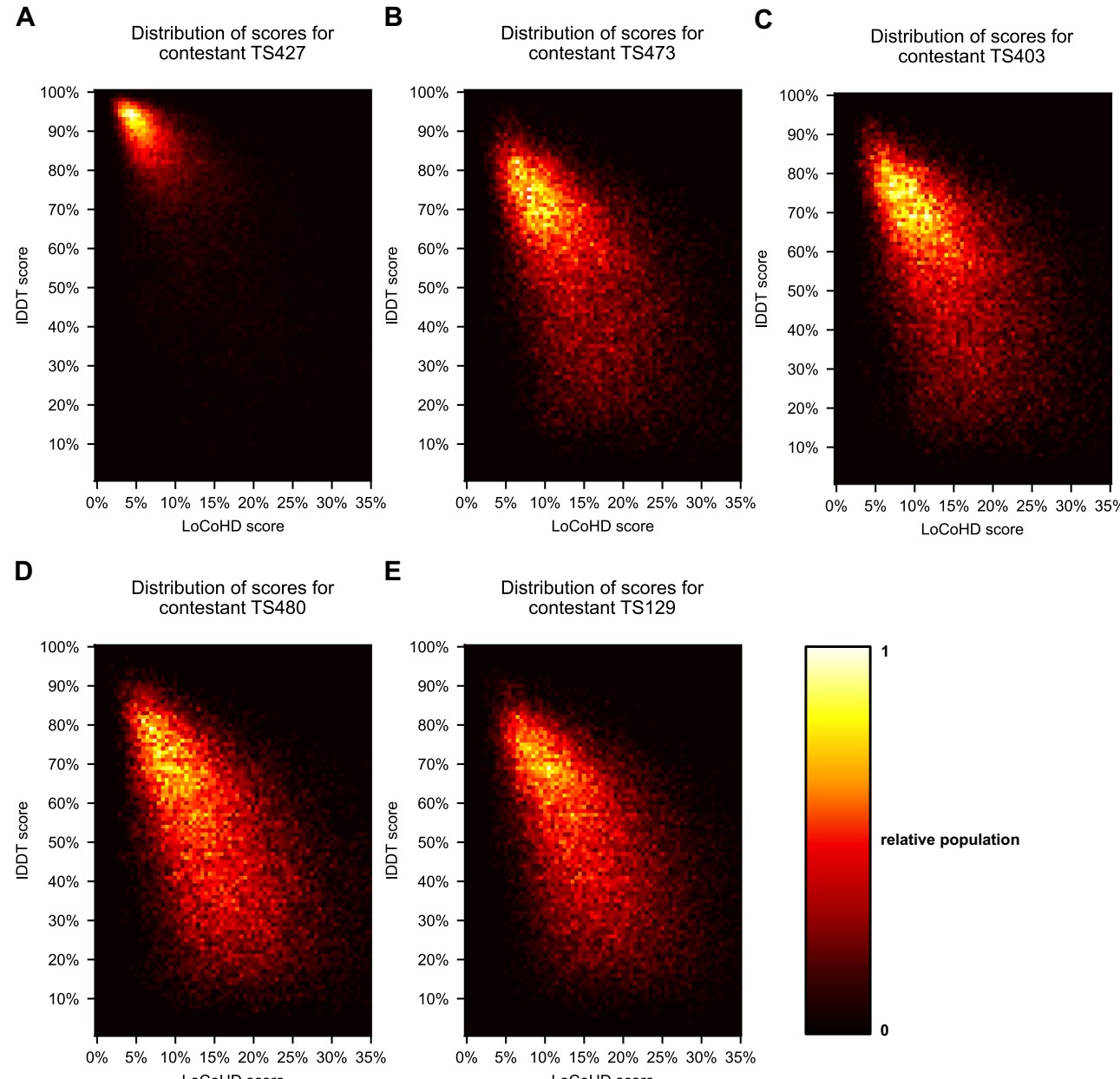

**Fig. 2 | The LoCoHD-lDDT relationship.** Visualizing the two-dimensional distributions of the per-residue LoCoHD-lDDT pairs for each CASP14 contestant examined. On panels **A**–**E** the histograms for the structure predictors AlphaFold2, BAKER, BAKER-experimental, FEIG-R2, and Zhang can be seen, respectively. Histograms are depicted as heat maps, with warmer colors indicating higher populations in the corresponding area. Source data are provided as a Source Data file.

The AlphaFold2 predicted structures T1064TS427_1 and T1064TS427_5 show large SpR, median LoCoHD, and median lDDT differences (first row, Supplementary Table 3). The experimental structure behind target T1064 is the SARS-CoV-2 ORF8 accessory protein (PDB ID: 7JTL[37]). Structures can be seen in Fig. 3, score-correlations, and per-residue LoCoHD scores can be seen in Supplementary Fig. 9. Although T1064TS427_1 has a much lower median LoCoHD than T1064TS427_5, the former structure has an outlier residue Lys[94] with an extremely high LoCoHD score at around 37% (residue numbering is according to the structure 7JTL) while this residue is by no means an outlier according to its lDDT score. In the experimentally determined structure, this lysine does not participate in interactions with any other residues and its Nζ faces the solvent bulk. In model T1064TS427_1, this lysine is H-bonded to the backbone of Leu[118], while being close to Tyr[79] and Phe[120], which increase the aromatic content around Lys[94]. In T1064TS427_5 the two aromatic

residues are farther apart than in the first structure, lowering the LoCoHD score to 27%. This high environmental difference is not reflected in the lDDT scores belonging to Lys[94] (37% in T1064TS427_1 and 38% in T1064TS427_5). The residue Glu[106] has the largest LoCoHD score in T1064TS427_5, but it does not appear in the top 10 residues with the largest LoCoHD scores in T1064TS427_1. In the experimental structure, Glu[106] participates in a relatively isolated H-bonding interaction with Tyr[38]. Although in T1064TS427_1 the two aforementioned residues are too far away for this H-bond to form, they are still close together, creating a similar environment around the glutamic acid. In T1064TS427_5 Glu[106] faces the solvent bulk, while Tyr[38] forms a π-π interaction with Tyr[105], which creates a highly different environment, than in the true structure. This example also shows that LoCoHD is able to differentiate chemically significant changes in protein structures that might be overlooked when using purely structure-based comparisons.

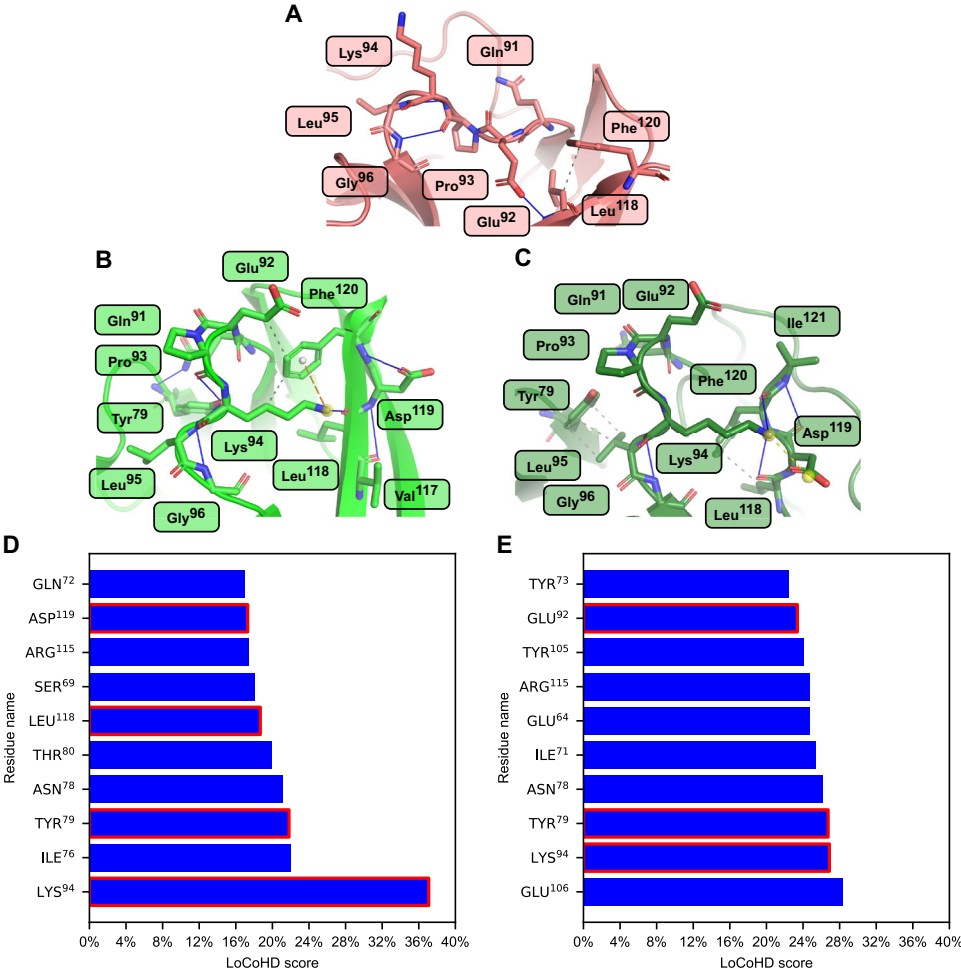

**Fig. 3 | Case study of the prediction target T1064.** Summarizing the residue environmental differences between the AlphaFold2 predicted structures T1064TS427_1 and _5 and the true, experimental SARS-CoV-2 ORF8 accessory protein structure (T1064). Panels **A**–**C** show the environments of the residue Lys[94] in the experimental structure, in T1064TS427_1 and in T1064TS427_5, respectively. Low opacity yellow spheres denote ionic interaction centers, yellow dashed lines denote ionic interactions, orange dashed lines denote pi-cation interactions, full blue lines denote H-bridges, gray spheres denote pi interaction centers and short-dashed gray lines denote van der Waals contacts. It can be seen that while in the true structure this lysine faces the solvent, in the predicted structures this residue participates in several inter-residue interactions. On panels **D** and **E** the first ten residues can be seen having the largest LoCoHD scores in T1064TS427_1 and in T1064TS427_5, respectively. For the residues that are present either on panel **B** or **C**, the bars are highlighted with a red contour. Source data are provided as a Source Data file.

Similar analysis was performed on some of the CASP15 contestants and their predicted structures. Global lDDT, LoCoHD and CAD-score[31] statistics for contestants TS229 (Yang-Server), TS278 (PEZY-Foldings), TS439 (Yang) and TS074 (DFolding) are presented in Supplementary Table 4 and in Supplementary Fig. 10. The former three contestants all achieve a median prm-lDDT of about 0.85, a performance similar to that of AlphaFold2 in CASP14. This performance is also reflected by the low median prm-LoCoHD values and again, the two scoring systems agree on the contestant order. Meanwhile, Supplementary Figs. 11–14 show the lDDT-LoCoHD analysis of two target structures and their predictions; H1166TS278_1 and _5, which is a human Fab S24-188 in complex with the N-terminal domain of the SARS-CoV-2 Nucleocapsid protein (to be published, PDB ID: 7SUE), and H1144TS278_1 and _5, which is a mouse/alpaca CNPase-Nb8d nanobody-antigen complex (to be published). The environmental differences in these complex structures, which were highlighted by LoCoHD, are chemically intuitive and are helpful in pointing out how a predictor weighs the relevance of the inter-residue interactions during the reconstruction of a complex.

**Comparison of structure ensembles through LoCoHD and RMSD**
For the analysis of the in-house determined NMR structure ensembles we chose the tryptophan cage fold extended by 5 residues, the so-called E5 miniprotein. The structural ensembles of this protein, each containing 50 different conformations, were previously determined at five different temperatures ranging from 277 K to 321 K[38]. With increasing temperature, the protein gradually loses its well-defined tertiary structure, which is reflected in the diversity of residue conformations (Fig. 4A, B). Supplementary Fig. 15A, B show the 50 by 50 LoCoHD distance matrices averaged over all primitive atoms and plotted as heatmaps. These scores were calculated using the FA typing scheme, resulting in 197 environment comparisons per structure pair. Each cell in this matrix describes the relationship of two structures within the ensemble, i.e. it is the average LoCoHD score of the primitive atom environments computed for the two structures in question. Meanwhile, Supplementary Fig. 15C, D shows the LoCoHD scores of each primitive atom. These scores are the averaged LoCoHD values over all structure comparisons. For the analysis of the E5 ensembles, these views convey orthogonal information, one about the overall

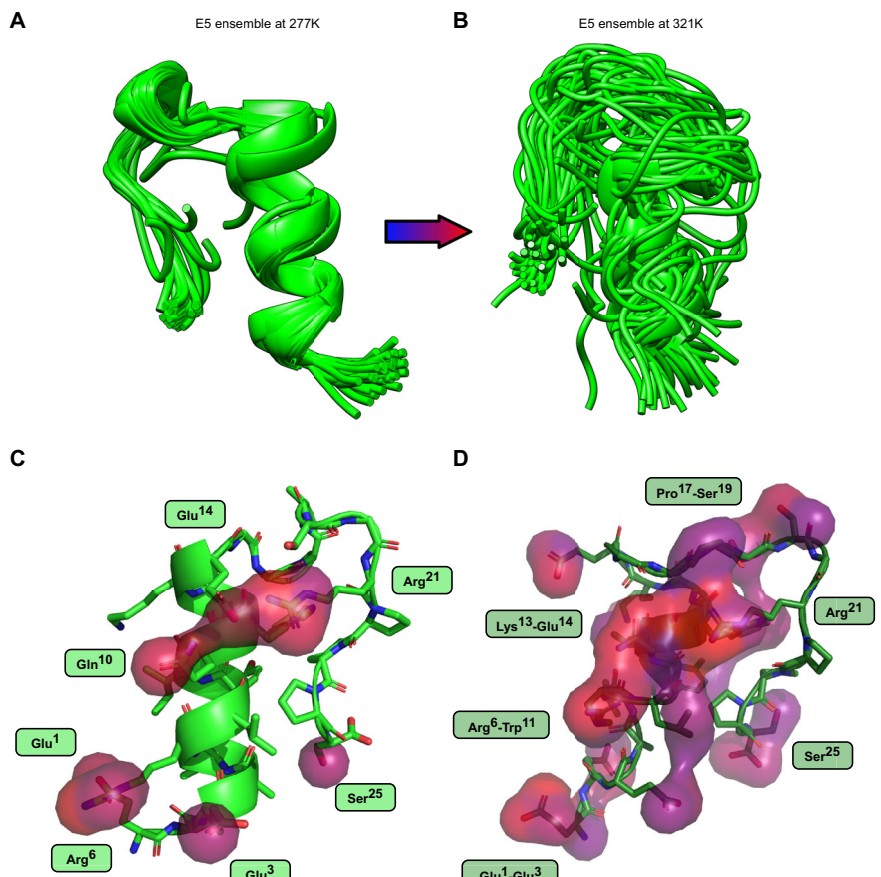

**Fig. 4 | Comparison of the structures within the E5 ensembles.** Panels **A** and **B** depict the structural ensembles of 50 conformers at temperatures of 277 K and 321 K, respectively. The E5 miniprotein consists of 25 residues, which fold into an N-terminal alpha helix, followed by a $3_{10}$ helix, and a C-terminal polyproline II helix. At higher temperatures these secondary structural elements disappear and the protein loses its well-defined structure. Panels **C** and **D** highlight the regions as surfaces within the 277 K and 321 K structures, respectively, where the ensemble-average LoCoHD scores of the primitive atoms are above 20%. In the low mobility "cold" structure only a few primitive atoms are present that reach this threshold. In contrast, in the high mobility "hot" structure a lot of high LoCoHD score regions are highlighted. Residues or residue-intervals owning these high LoCoHD primitive atoms are also indicated in boxes. Surfaces are colored according to primitive atom LoCoHD scores, with warmer colors indicating higher values (from blue, through purple, to red). Source data are provided as a Source Data file.

dissimilarity of the ensemble elements (different conformations), and one about the environmental variability of the primitive atoms.

At 277 K the structure of E5 is well ordered. The overall environmental variability of the ensemble is low, as only 3-4 structures are significantly different from the others (i.e. structures with approximately 15-17% LoCoHD away from the other structures), and the median LoCoHD of the primitive atoms is 7.4%. These can be observed on the corresponding distance matrix (Supplementary Fig. 15A), where only the bottom rows and leftmost columns are shown in warm colors, and on the corresponding growth plot (Supplementary Fig. 15C), where most of the values are between 3.7% and 14.5%. In contrast, at 321 K the miniprotein shows high disorder and a broad conformational distribution. The distance matrix of this ensemble (Supplementary Fig. 15B) contains high LoCoHD values, mostly above 15%, with only about 5 structures showing some similarity (upper left blue patch). The median LoCoHD of the primitive atoms (Supplementary Fig. 15D) also shifted from 7.4% to 15.1%. This leaves the one sigma range of primitive atoms between 11.5% and 21.3% at 321 K.

The scatter-plots of the LoCoHD-RMSD pairs are depicted in Supplementary Fig. 15E, F for ensembles at 277 K and 321 K, respectively. In these plots, each point belongs to a structure-structure comparison, resulting in 1225 points per plot. The LoCoHD and primitive atom RMSD values show high correlation at all temperatures in the E5 ensembles. SpR values range from 0.74 (at 310 K) to 0.88 (at 277 K) with no obvious connection between the SpR and the temperature of the ensemble. At low temperatures, these points form visually three (277 K) or two (288 K and 299 K) clusters, while at higher temperatures this behavior is not observed, leaving only one point cloud. The separation of these clusters mainly happens due to the RMSD metric, since it produces obvious boundaries, while the LoCoHD scores do not distinguish such sharp separating values in these cases. For example, at 277 K there are three "visually obvious" RMSD boundaries at approximately 2 Å, 3 Å and 3.75 Å. Performing agglomerative clustering using the RMSD matrix (with a distance threshold of 2 Å and complete linkage) results in five clusters. The first cluster consists of 46 structures and the remaining structures form clusters by themselves, indicating that these are outlier structures. However, despite the absence of an obvious LoCoHD boundary, when the same clustering procedure is applied on the LoCoHD matrix, but with the number of clusters set to five, the same outlier structures are identified. Also, when the LoCoHD-RMSD point cloud at 321 K is inspected, it can be noted that between the narrow range of LoCoHD scores of 17.1% and 18.7%, the points have a large spread along the RMSD axis (StDev = 1.0 Å and a maximum distance difference of 5.6 Å). This indicates the existence of a large structural difference-range within a small environmental composition difference-range.

An NMR-derived ensemble of the W316A, M317A mutant Gag-Pol polyprotein of HIV-1 (between residues $Pro^{133}$-$Val^{353}$) was downloaded from the PED (PED ID: PED00072e000, residues numbered by UniProt ID: P12493, PDB ID: 2M8P) and subjected to similar analyses[39]. These

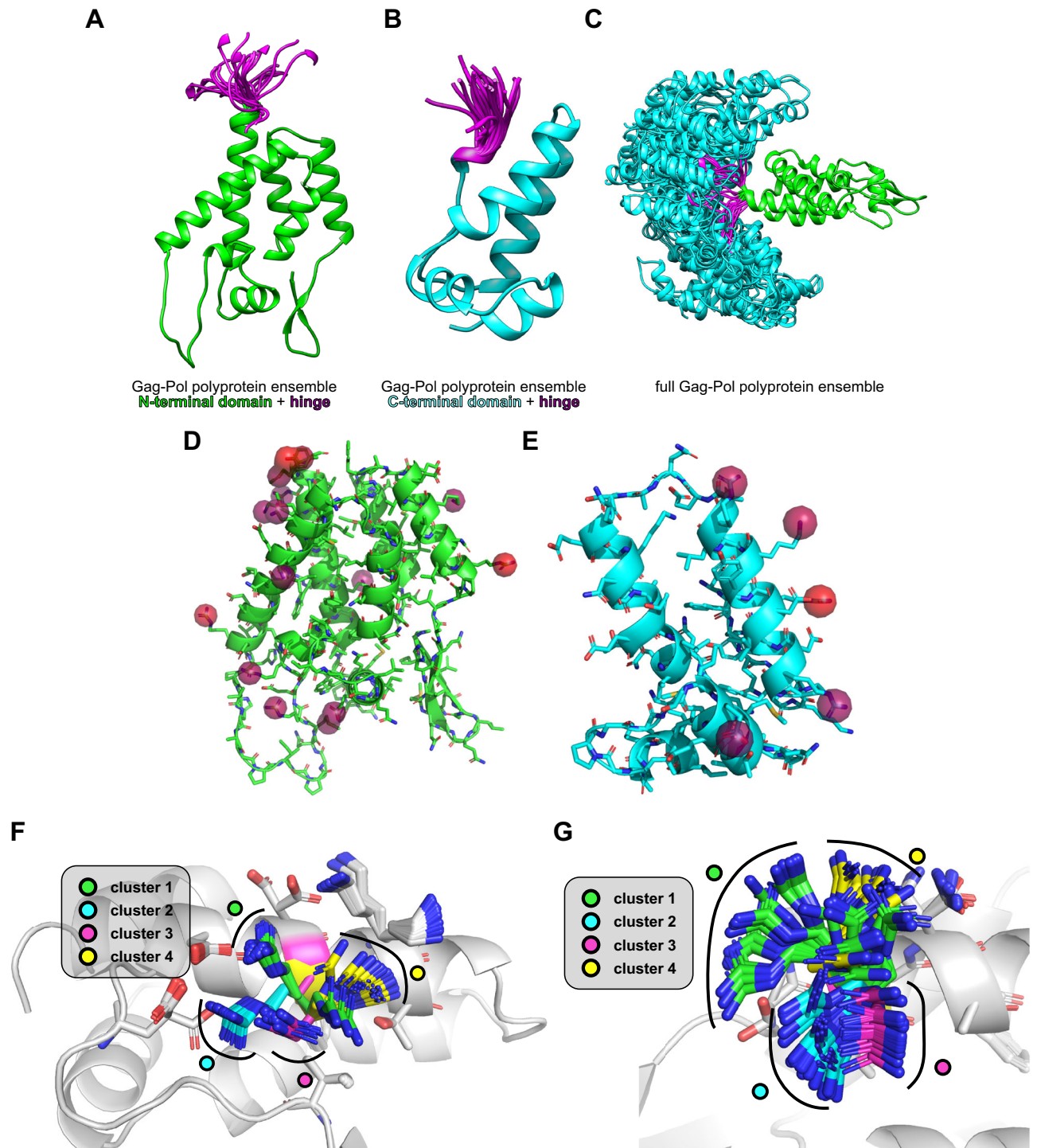

**Fig. 5 | Case study using the HIV-1 Gag-Pol polyprotein.** Panels **A**–**C** show the N- and C-terminal domains, and the full ensemble of the HIV-1 Gag-Pol polyprotein, respectively. Both panels show an ensemble of aligned structures. It can be seen, that while the domains themselves are very rigid, the hinge region connecting them (colored magenta) is highly mobile. Panels **D** and **E** show the regions in the N- and C-terminal domains with the highest (greater than 20%) average LoCoHD scores highlighted as semitransparent surfaces. These surfaces are also colored according to the average LoCoHD score in that region. The relatively low number of such primitive atoms indicate low chemical variability. Panels **F** and **G** show realizations of LoCoHD clusterings with a cutoff value of 7%. These panels focus on residues $Arg^{299}$ and $Arg^{150}$, respectively, along with their environments, which can (at least partially) explain the separation between the four clusters. Source data are provided as a Source Data file.

structures consist of two rigid domains, connected by a flexible hinge region between residues $Ser^{278}$-$Leu^{283}$ as depicted in Fig. 5A, B. The structural diversity of this ensemble is mainly due to the different conformational states of this hinge region (Fig. 5C). In ensembles like this, the RMSD after an all-atom alignment usually produces large and uninformative values, since a global alignment cannot be performed optimally on the two domains at the same time.

For the Gag-Pol polyprotein, we used the CG typing scheme instead of the FA scheme, due to the large number of atoms present in the system and the resulting high computational time. Also, rather

than comparing the full ensemble of 100 structures, we only compared the first 50 structures in the ensemble. The molecular surface colored according to the LoCoHD values is depicted in Supplementary Fig. 16A, while the first twelve primitive atoms with the largest LoCoHD values are listed in Supplementary Table 5. The structure-structure LoCoHD distance matrix was also calculated and is shown in Supplementary Fig. 16B, along with the ordered primitive atom LoCoHD values in Supplementary Fig. 16C. When compared with the distance matrices of E5 (Supplementary Fig. 15A, B), a different pattern emerges. Blocks with low LoCoHD scores can be easily distinguished, suggesting high structural clusterability. This can also be seen in the LoCoHD - RMSD scatter-plot (Supplementary Fig. 16D), where an obvious gap can be seen between low and high LoCoHD values, separating the points into two clusters. In this case the RMSD values (calculated for all structure-structure pairs) are between 0 Å and 20 Å and do not discriminate clusters. The SpR between the LoCoHD values and the RMSD values is 0.24, which is lower than for E5 at any temperature. When agglomerative clustering is applied to the LoCoHD distance matrix with a distance threshold of 7% (a value in the gap between the two LoCoHD score clusters), 4 distinct clusters are produced. Performing the clustering on the RMSD matrix with a cluster number of 4 does not produce the same clusters. Again, this is in contrast to the case of E5, where clustering on the LoCoHD and RMSD matrices produced the same result. Based on the LoCoHD analysis, we can conclude that the fluctuation in relative domain positions causes little change in the chemical environment within each domain (Fig. 5D, E). The largest changes in LoCoHD indicate that, with the exception of the N-terminal 2 residues (Tyr277, Ser278) of the loop connecting the two domains, the structure is not perturbed by the domain movements - the most significant chemical changes within the domains are observed by three Arg residues occupying different niches as they rotate on the surface of the protein (Fig. 5F, G), independent of the large domain fluctuations.

Interestingly, one of these, Arg150 (or Arg18, according to a different numbering convention), was shown crucial for the formation of the hexameric capsid of HIV-1[40,41,42]. Mutations at this site result in distinct morphological variation of the viral assembly without causing conformational changes discernible by solid state NMR. LoCoHD identified this residue as being able to detect conformational fluctuations of the matrix – as would be expected of a residue that recognizes the presence of interaction partners and guides the assembly process.

An additional ensemble LoCoHD-RMSD comparison analysis can be found in Supplementary Note 5 and in Supplementary Fig. 17.

## Using LoCoHD for the analysis of an MD simulation

Molecular dynamics simulations generate trajectories of proteins or protein complexes that represent the conformational changes of the systems under study. These trajectories are hundreds of thousands of time-correlated samples from a structural ensemble. Since a thorough visual inspection of these trajectories is problematic due to the size of these datasets, several numerical tools have been developed to plot the time dependence of different descriptors (such as RMSD, solvent accessible surface area, principal components, etc.). Here, by analyzing the MD trajectory of the dimeric form of a structural protein of the renal filtration barrier, podocin (UniProt ID: Q9NP85)[43], we show that the time-dependent LoCoHD score of different residues can pinpoint structurally important changes of the simulated protein. The CG+Cent typing scheme was used with the uniform weight function between 3 Å and 10 Å and the residue centroid primitive atoms as anchor atoms. The trajectory was analyzed between 600 ns and 1600 ns with 2.5 ns intervals. Each frame was compared to the first frame at 600 ns anchor atom by anchor atom, and the time dependence of the LoCoHD scores was recorded.

After visual inspection of the LoCoHD score vs. time plots, it became clear that some residues fluctuated between two or more different environmental compositions and arrangements. To objectively select these residues from the 344-residue long homodimer, we calculated Sarle's bimodality coefficient (denoted by β)[44] for the LoCoHD distribution of each residue;

$$\beta = \frac{\gamma^2 + 1}{k}$$

where γ is the sample skewness of the distribution and κ is the sample kurtosis. The higher this number is, the more likely it is that the distribution of these scores is bimodal. The value β = 0.555 is a good reference, since it belongs to the uniform distribution. Any distribution above β = 0.555 is likely to be bimodal. Using this procedure, we identified six residues - His276 (chain A, β = 0.75), Gly273 (chain B, β = 0.66), Met197 (chain A, β = 0.65), Asp267 (chain B, β = 0.64), His276 (chain B, β = 0.63), and Phe176 (chain B, β = 0.63) - as the residues with the most bimodal LoCoHD score distributions. The LoCoHD score vs. time plots for these residues are depicted in Supplementary Fig. 18.

The two environmental states of the residue His276 can be easily characterized by the $X_1$ angle (N-Cα-Cβ-Cγ dihedral angle) of the histidine. This dihedral angle takes on values from two angle-ranges, one between (+155°, −165°) (minor form), and one between (+40°, +90°) (major form). When His276 takes on the former conformation, the histidine side chain faces the bulk solvent and allows the backbone carbonyl group of Gly273 - another highly bimodal residue - to form a H-bond with the backbone NH group of Ser277, extending a short α-helix (Fig. 6A, B). However, when His276 is in its major form, it is positioned between Gly273 and Ser277, blocking the formation of the aforementioned H-bond and shortening the helix. This behavior is more dominant in the case of His276 in chain A. In the case of His276 in chain B, the histidine also moves away from the Gly273-Ser277 pair, but the formation of the Gly273-Ser277 backbone H-bond appears to be more sporadic than in the case of chain A.

In the case of the residue Met197 in chain A, the two states are realized by the orientation of the Met side chain with respect to a hydrophobic core (Fig. 6C, D). In one setting, the Cε atom stays close to Val165, Asp166, Leu167, Asn199, and Ala200. This state is similar to the starting state (t = 600 ns) and has a low LoCoHD score (-8-10%). At about 1000 ns, the methionine side chain moves away from these residues and fills a previously unoccupied hydrophobic cavity, created by Val165, Gln170, Tyr195, Leu203, Val210, and Ile258. This state is different from the initial state and has higher LoCoHD scores (-16-20%). Here, the saturability of environments can be easily observed, when the LoCoHD time dependence of Ala200 and Leu203 are inspected (Fig. 6E, F). Ala200 has relatively few neighbors besides Met197, and thus the same sharp change in LoCoHD score can be observed at 1000 ns, since its environmental composition is mainly determined by the primitive atoms from the methionine. In contrast, Leu203 is constantly surrounded by other residues (Leu204, Leu205, Leu207, Val210, Ile258), and although Met197 comes close to it at 1000 ns, no visible correlation between the two LoCoHD score time dependencies can be observed.

The 273–277 segment is in the critical hinge region of the podocin monomer, the flexibility of which was suggested to influence the effect that pathological mutations exert[43]. Thus, recognizing that two interaction-wise different orientations are sampled by His276 may carry functional significance.

## Discussion

The in silico study of protein structures requires precise mathematical and computational techniques capable of distinguishing between different conformational states and residue-residue interaction network topologies. Existing methods that aim to measure differences between atomic arrangements focus mainly on atomic coordinates or interatomic distances, ignoring important physicochemical differences between the states under study. To overcome these limitations, we

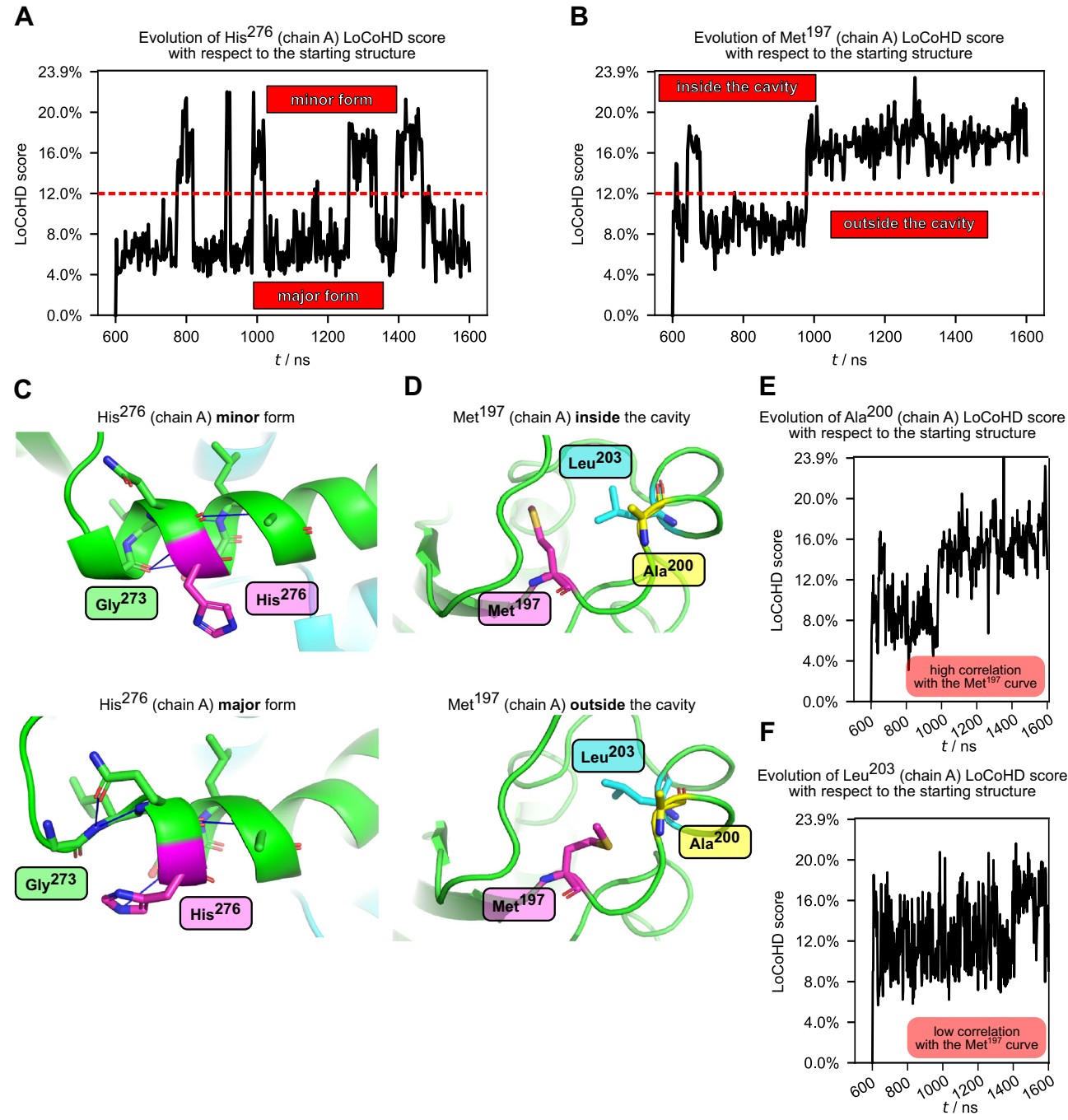

**Fig. 6 | Use of LoCoHD for an MD analysis.** Panel **A** shows the time dependence of the LoCoHD score of the highly bimodal residue His[276], while also showing the clear separation of the two modes. Panel **C** shows the atomic representation of these modes, distinguishing a long (minor) and a short (major) helical form. Full blue lines denote H-bridges. The important interacting partners are highlighted by sticks. His[276] is shown in purple. Panels **B** and **D** show the same representations for Met[197]. Here the two forms are the occupied and empty cavity forms. The system starts with the methionine sidechain outside the cavity and then fills the cavity at about 1000 ns due to the conformational change of the aforementioned sidechain. This behavior can also be followed from the time dependence of the LoCoHD score of the initially close residue Ala[200] in panel **E**. In contrast, in the case of Leu[203], which also becomes close to Met[197] after filling the cavity, this correlation is not present due to the environmental saturation effect (panel **F**). Source data are provided as a Source Data file.

have developed the local composition Hellinger distance (LoCoHD) metric and demonstrated that it is able to discriminate between different chemical compositions of residue environments. This metric is highly flexible, offering several customizable points in its workflow, and opens up an unexplored area of composition-based residue environment investigations. Specification of the task is completed in three conceptually separate steps: the choice of anchor atoms provides global spatial resolution and helps focusing on the region of interest; the choice of weighting scheme provides local spatial resolution, allowing the specification of what the user considers to be the environment of an anchor atom; and the choice of the primitive typing scheme provides the chemical resolution, a chance to specify which atoms (or atom groups) should be differentiated based on their nature.

First, we were able to assert the environmental similarity of all amino acid type pairs between randomly selected and uncorrelated residue environments. In this way, it was possible to construct the full

distribution of LoCoHD scores, which provided comparative information with other LoCoHD measurements. It is important to note that this distribution is dependent on the primitive typing scheme and weight function and should be recalculated for each new setup. Nevertheless, for the two different typing schemes used for this task, we observed a β distribution of LoCoHD values in both cases. These had averages of 23.92% and 26.83%, providing good benchmarks for deciding whether a score is considered large or small. The average LoCoHD values of specific residue type and category pairs were also compared to these global values and to each other. These results respected the chemical intuition regarding the behavior of amino acids exceptionally well.

Secondly, the correlation between the LoCoHD score and the lDDT score, and the correlation between the LoCoHD score and the RMSD score were also examined. While these correlations were high for protein pairs with high structural similarity, the different information content of these descriptors became more apparent as the structures became more dissimilar. LoCoHD was able to rank the top five CASP14 competitors in the same order as lDDT. However, the absolute correlation between the residue lDDT scores and the residue LoCoHD scores gradually decreased as the predictive power of the competitors decreased. In the case of the RMSD score, protein ensembles with high internal structure-structure similarity showed a high LoCoHD-RMSD absolute correlation, but this absolute correlation also decreased with lower internal similarities, as in the case of multi-domain proteins or IDPs.

Finally, we demonstrated the use of the LoCoHD score in a molecular dynamics setup. This descriptor, used in a time-dependent manner, was able to pinpoint structurally important residues within the podocin dimer simulation. Highly bimodal LoCoHD score distributions corresponded to bimodal environmental states. We propose that inspection of time-dependent LoCoHD graphs can suggest trajectory convergence, highlight regions where residues undergo interaction mode changes, or - when compared to different energetically optimal environmental arrangements - even provide a distance measure from local optima.

## Methods
### Description of the LoCoHD algorithm
To characterize local chemical differences by calculating LoCoHD scores, two protein structures must be provided, which are then treated as labeled point clouds. In theory, this initial labeling can contain as much information as desired, but during the development and testing of LoCoHD we simply considered atoms to be centers of interest and labeled them with their standard PDB name and the name of the source residue to which the atom belongs. Next, the initial point clouds of both proteins are mapped to new point clouds, for which the new labels are chosen from a finite set called the "primitive type set". These primitive types should preferably contain chemical quality descriptive information. For example, one may map the glutamic-acid Oε1 and Oε2 atoms to the negative oxygen primitive type (O_neg), while the serine Oδ atom to the neutral oxygen primitive type (O_neu), discriminating the two chemically different oxygen-atom types. During the mapping from the initial atom cloud to the primitive atom cloud, any number of atoms can be omitted. Thus, in all of our calculations we only considered heavy atoms and ignored all H-atoms. Furthermore, virtual sites can also be introduced into the primitive atom cloud, like specific atom-set centroids or center-of-mass sites, with their own designated primitive types.

Once the primitive atom clouds have been created, a certain subset of primitive atoms must be selected from both clouds in such a way that for each primitive atom in one subset must have at least one corresponding primitive atom in the other. These atoms are called "anchor" atoms and they form the basis of the LoCoHD comparisons. For each corresponding anchor atom pair, our algorithm computes a

LoCoHD score, which reflects the difference in the primitive type composition between the environments of the anchor atoms. Since the selection of the anchor atoms defines not only the global spatial resolution and the focus area of the comparison but also its resource-efficiency, the applied anchoring-scheme has to be adapted to the task at hand.

The LoCoHD score for a given anchor atom pair $(i, j)$ is calculated as follows:

$$\mathrm{LoCoHD}_{ij} = \int_0^\infty w(r)\, H(\boldsymbol{\Phi}_i(r), \boldsymbol{\Phi}_j(r))\, \mathrm{d}r$$

where $w(r) \geq 0$ is a weight function whose integral on $(0, \infty)$ is 1, $H$ is the Hellinger distance between the two probability mass functions (PMFs), $\boldsymbol{\Phi}_i(r)$ and $\boldsymbol{\Phi}_j(r)$, and $\boldsymbol{\Phi}_i(r)$ is the distance-dependent environmental composition (DDEC) of the $i$th anchor atom: a vector with positive entries and an L1 norm of 1, and it has as many dimensions as many primitive atom types are used. $\boldsymbol{\Phi}_i(r)$ contains the fraction of occurrence of each primitive type within a sphere around the $i$th anchor atom with a radius of $r$.

As an example, suppose a primitive type set of {A, B, C} is used and the environment of the anchor atom is described by the set {(A, 0 Å), (A, 1 Å), (B, 3 Å), (B, 5 Å)}, in which each entry is a (primitive type, distance from anchor atom) pair. This means that the anchor atom, which has a primitive type of 'A', is surrounded by 3 other primitive atoms. $\boldsymbol{\Phi}(r = 2\,\text{Å})$ in this case would be (1, 0, 0), since only the primitive type 'A' is present in a 2 Å sphere around the anchor, while $\boldsymbol{\Phi}(r = 4\,\text{Å})$ would be (0.66, 0.33, 0) (two 'A' and one 'B' type inside the sphere), and $\boldsymbol{\Phi}(r = 7\,\text{Å})$ would be (0.5, 0.5, 0).

The Hellinger distance[45] of two PMFs (here, **p** and **q**) is given by:

$$H(\mathbf{p}, \mathbf{q}) = \sqrt{\frac{1}{2} \sum_i (\sqrt{p_i} - \sqrt{q_i})^2}$$

which guarantees a result between 0 and 1, with 0 meaning total PMF similarity, and 1 meaning total PMF dissimilarity. Since the weight function is chosen so that it satisfies the properties of a probability density function (PDF), the LoCoHD integral is a weighted average of Hellinger distances, also resulting in a value between 0 and 1.

Due to the discrete nature of atomic positions, the Hellinger distance between the two DDEC functions is constant for specific $(r_n, r_{n+1})$ intervals, with values denoted by $H_n$. This means, that the LoCoHD integral can be simplified into an easily computable form:

$$\mathrm{LoCoHD} = \sum_{n=0}^{N-1} H_n \int_{r_n}^{r_{n+1}} w(r)\, \mathrm{d}r = \sum_{n=0}^{N-1} H_n (W_{n+1} - W_n)$$

Here, we omitted the indices $i$ and $j$ of the anchor atoms for the purpose of readability. In these equations $r_0$ is considered 0 Å, while $r_N$ is considered $\infty$ Å. The values $W_n$ are the antiderivatives of $w(r)$ evaluated at $r_n$.

In addition to omitting all initial (i.e. non-primitive) atoms, it is also possible to omit primitive atom types from an environment depending on the central, anchor atom. This is an important feature, since an atom from a particular amino acid is always surrounded by the other atoms from the same amino acid. For small distances this will make the DDEC functions more similar, i.e. this will add a systematic, residue-type dependent bias into the LoCoHD scores. Therefore, it is advantageous to ignore primitive atoms that belong to the same residue as the anchor atom. This feature is referred to as "using only hetero-residue contacts".

From the previous example, the environment can be expanded with residue-source information; the original set becomes {(A, 0 Å, X), (A, 1 Å, X), (B, 3 Å, Y), (B, 5 Å, Z)}, where each third component (X, X, Y, and Z) denotes the residue-source. Since the anchor atom (A, 0 Å, X) is

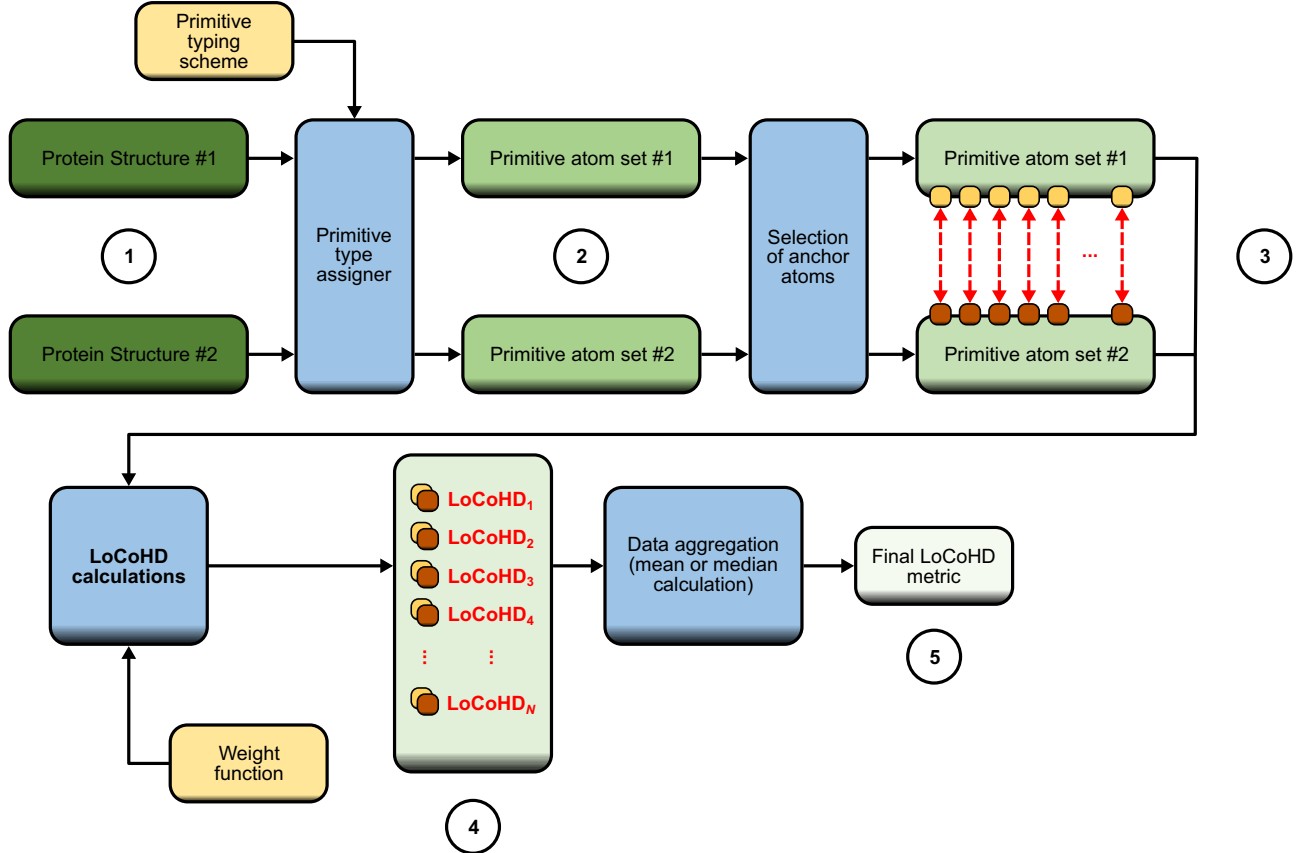

**Fig. 7 | General flowchart for the LoCoHD algorithm.** Starting from the protein structures, the procedure first maps the set of real atomic coordinates and names (stage **1**) to primitive atoms (stage **2**). How this mapping is done can be set through the primitive typing scheme. Then, a subset of these primitive atoms is selected as anchor atoms (light and dark brown squares, stage **3**). The figure emphasizes the need for a surjective correspondence between these anchor atoms (red dashed arrows). For each pair of anchor atoms a LoCoHD calculation is performed and the results of the environment comparisons are obtained (stage **4**). The LoCoHD calculations are dependent on the shape of the weight function employed. One can use the set of these LoCoHD scores directly, or perform a final average (or median) calculation that yields a single number describing the structural similarity of the two protein structures (stage **5**).

from the residue X, the primitive atom (A, 1 Å, X) would be omitted from the environment if only hetero-residue contacts are allowed.

The whole procedure is depicted in Fig. 7 and in Supplementary Fig. 19.

## Primitive typing schemes

During our work, we investigated several different primitive typing schemes. These schemes can be characterized by either being "full atom" (FA) or "coarse grained" (CG) in nature, or whether they contain residue centroid primitive atoms or not (Cent: a virtual atom at the geometric center of every residue). These different primitive typing schemes are referred to as FA, CG, FA+Cent and CG+Cent.

In the FA typing scheme, all heavy atoms in the original structures are mapped to primitive atoms. The primitive types of these atoms are assigned from the following primitive type set: negative oxygen (O_neg), neutral oxygen (O_neu), positive nitrogen (N_pos), neutral nitrogen (N_neu), aliphatic carbon (C_ali), aromatic carbon (C_aro), and sulfur (S). In FA+Cent an additional residue centroid primitive atom is used, with a primitive type of Cent, and coordinates set by the geometric center of the residue's heavy atoms. Thus, the DDEC functions produce 7-dimensional vectors in FA, and 8-dimensional vectors in FA+Cent.

In the CG typing scheme the following primitive types are distinguished: amide group carbon atoms (AmideC), alcoholic OH group oxygen atoms (OH), positive centers (Pos), negative centers (Neg), aromatic centers (Aro), aliphatic centers (Ali), and sulfur atoms (S).

Note, that some of these primitive atoms are not mapped from one atom, but rather from the geometric centers of certain atom-groups. Examples are the Oδ1-Oδ2 atom group of Asp for a negative center, or the Cγ-Cδ1-Cδ2-Nε1-Cε2 atom group of Trp for an aromatic center. The CG+Cent typing scheme also contains the heavy atom centroids of the residues, similar to FA+Cent.

Schemes FA and CG are useful when a one-to-one correspondence can be established between each atom of the two structures, i.e. all resulting primitive atoms can be used as anchor atoms. This is the case when comparing different conformations of the same protein (as in the case of an NMR ensemble or the trajectory of a molecular dynamics simulation), or when comparing the experimental structure of a protein with its predicted structure (as in the case of CASP competitions). In contrast, the FA+Cent and CG+Cent typing schemes are useful when the two proteins to be compared do not have the same primary structure and thus contain different residues and atoms. In these cases, the Cent primitive atoms can serve as anchors, through which the LoCoHD calculations are performed.

Selection of the primitive atoms is again, task dependent. The FA and FA+Cent schemes provide the most chemical resolution. In the case of FA and CG, anchor pairing is only trivial if the two structures to be compared are comprised of the same atoms. Centroid-containing schemes (like FA+Cent and CG+Cent) can be used if residue-sized global spatial resolution suffices, and they also offer a way to reduce the runtime of the metric calculation.

## Random LoCoHD distribution generation

To generate these experimental distributions, we used a homology-filtered PDB database from PISCES[46], culled on 2022.02.22. We used a maximum sequence identity of 25%, a resolution of 2 Å, an R-value of 0.25, and a protein chain length of 300 residues. This resulted in a database with a total of 3444 pdb files. The order of these pdb files was shuffled, and successive pairs of structures were considered using the shuffled order. For each pair of structures, all residues of the structure with the smaller number of residues were randomly paired with residues of the other structure. In this way, we were able to generate random residue pairs with uncorrelated environments. The LoCoHD values of these pairs were then calculated using the FA+Cent and CG+Cent typing schemes and the uniform weighting function between 3 Å and 10 Å. Only hetero-residue contacts were taken into account.

## Construction of the LoCoHD predictor neural network

The neural network was constructed, compiled, trained and evaluated using the Python3 TensorFlow 2.15.0[47] package. A Siamese architecture was used. The network required a pair of 26-dimensional vectors as inputs, which can be split into two parts: a 20-dimensional one-hot encoded vector, designating the central residue type, and a 6-dimensional interaction-count vector, counting the number of interactions for each interaction type the central residue makes (H-bonds, van der Waals interactions, disulfide bonds, salt bridges, π-π stacking and π-cation interactions). These interactions were identified using the RING standalone software[48] and counted using in-house Python3 scripts. The network first creates internal representations of these vectors through a weight-shared feedforward 2-layered arm-pair with layer sizes of 256 and 128 neurons and ReLU activations. The resulting 128 dimensional vectors are then subtracted from each other and their difference is squared, resulting in a single 128 dimensional vector. Note, that this intermediate result is invariant with respect to the order of the inputs, making the network symmetric. Then, a simple, 3-layered feedforward network processes this further with layer sizes of 128, 64 and 1 neurons, and activations of ReLu, ReLu and sigmoid, respectively. The total number of learnable network parameters came to be 64641. Weight initialization was performed with the uniform Glorot initializer. Training was performed with the Adam optimizer (learning rate = 0.001) and the binary crossentropy loss (since the output can be thought of as a fuzzy binary categorization). Metrics of mean squared error and mean absolute error were used. Training was done on batch sizes of 64 for 3 epochs and with a validation split of 20/80. A total of 409408 environment-pairs were used for training obtained from the random LoCoHD distribution generation.

## CASP naming convention

In CASP[49–51], each contestant and target structure has an identifier code in the form of TS[contestant ID] and T[target ID], respectively. Targets are sometimes prefixed with H, denoting heteromer target prediction, instead of T, denoting tertiary-structure target prediction. In addition, each contestant provides five predicted structures, which are denoted by the codes T[target ID]TS[contestant ID]_[structure ID]. An example for this kind of notion is T1026TS473_2, which is the second prediction of TS473 (the predictor named BAKER) for the target T1026 (FBNSV capsid protein, PDB ID: 6S44). We used this naming convention when referring to CASP protein structures.

## Processing and score calculation for the CASP structures

Structures were downloaded as tar-files from the CASP archive and were preprocessed with in-house Python3 scripts. Briefly, all experimental and predicted structures were loaded into memory with Bio-Python 1.81[52], non-canonical amino acids and disordered elements were removed from reference structures, correct chain-name-pairing was sought using sequence alignment based on the experimental

chains, and residues and atoms were removed from predicted structures that were not present in the experimental ones. Next, using all remaining atoms in the predicted structures, they were compared to the experimental structures using the lDDT and CAD score calculation modules of OpenStructure 2.7.0[53]. These per-residue scores were further extended with our LoCoHD calculations.

## Ensemble analyses

To investigate the differences between distance-matrix based LoCoHD calculations and alignment based RMSD calculations, we compared different conformations of protein structures within ensembles using both methods. These ensembles were obtained from previously published in-house NMR measurements[38] and also from the Protein Ensemble Database (PED)[54]. The LoCoHD calculations were performed using the FA and CG (centroid-less) typing schemes and the uniform weight function between 3 Å and 10 Å, allowing hetero-residue contacts only. Since an ensemble contains different conformations of the same protein, all primitive atoms were used as anchor atoms. The atomic coordinates of the primitive atom sets were used for the calculation of the optimal rotation matrix in the singular value decomposition (SVD)-based alignment algorithm and also for the calculation of the RMSD values. Within an ensemble, each structure was compared to every other with respect to their primitive atoms, resulting in a total of $M * N * (N-1)/2$ comparisons, where $M$ is the number of primitive atoms inside the protein and $N$ is the number of structures within the ensemble. In other words, a symmetric, zero diagonal, $N$ by $N$ distance matrix was obtained for each primitive atom.

## Visualizations

Structural visualizations were done using PyMol[5] 2.5.0, while PLIP[55] 2.3.0 was used for the visualization of residue-residue interactions. Graphs and plots were created with Matplotlib[56] 3.8.2.

## Statistics and reproducibility

Sample sizes in the PISCES dataset analysis were determined by the homology filtering method and the random residue pairing method detailed above. The reported statistical descriptors correspond to a single randomized run on all structures included in this study. No blinding was used for the assessment of the outcomes.

## Reporting summary

Further information on research design is available in the Nature Portfolio Reporting Summary linked to this article.

# Data availability

The datasets generated, analyzed and necessary for the reproduction of the case studies (except for the restricted part of the CASP15 dataset) are collected and available in a Figshare repository[57] with an accession code of https://doi.org/10.6084/m9.figshare.24885540. Besides the repository, all structural datasets used in this paper are also freely downloadable from the CASP database (https://predictioncenter.org/download_area/), from RCSB PDB (https://www.rcsb.org), from PED (https://proteinensemble.org), or from PISCES. The podocin MD trajectory and PDB accession code lists used in this study are also contained within the Figshare repository. Some of the CASP15 structures (and data related to them) are still under embargo by their authors' request and must be requested from the the CASP15 organizers at casp@predictioncenter.org. Specifically, in this manuscript, the CASP15 structures publicly available on 2023.12.31. were used, in addition to the restricted targets for invitees requested on 2024.02.16. The release of the embargoed data can be followed at https://predictioncenter.org/casp15/targetlist.cgi (Description column). Source data for the figures and tables are provided with this paper. Source data are provided with this paper.

## Code availability

The Rust and Python code for the LoCoHD project, along with dependency descriptions[52,58] and the Python scripts of the case studies are all available at the GitHub repository https://github.com/fazekaszs/loco_hd. A release version of v0.1.4[59] was used for this study. Brief details for the implementation can be found in Supplementary Note 6.

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

## Acknowledgements

This work was completed in the ELTE Thematic Excellence Programme supported by the Hungarian Ministry for Innovation and Technology. Project no. 2018-1.2.1-NKP-2018-00005 has been implemented with the support provided from the National Research, Development and Innovation Fund of Hungary, financed under the 2018-1.2.1-NKP funding scheme. Project number RRF-2.3.1-21-2022-00015 is implemented with the support of the European Union's Recovery and Resilience Instrument. Supported by the Ministry for Innovation and Technology from the Hungarian NRDI Fund (2020-1.1.6-JÖVŐ-2021-00010). All funding were awarded to A.P.

## Author contributions

The manuscript was written through contributions of all authors. All authors have given approval to the final version of the manuscript. Zs.F. proposed the project, designed the algorithms and experiments, implemented the LoCoHD software, collected data, conducted the computations and analyzed the results. D.K.M. and A.P. helped with experiment design, analysis and manuscript proofing. A.P. supervised the project.

## Funding

## Competing interests

The authors declare no competing interests.
