## [Peer Review File · Nature Communications]

LoCoHD: a Metric for Comparing Local Environments Of ProteinsReviewer #1 (Remarks to the Author):

The authors developed a metric (LoCoHD) for the highly flexible local composition Hellinger distance, which is based on the chemical composition of local residue environments. Distinguishing between different conformational states remains a challenge, particularly in the protein structure prediction, where it can be difficult to differentiate whether predicted structures are erroneous predictions or simply correspond to different conformational states. Overall, I am quite interested in this work. There are some questions needed to be addressed. See my comments below,

- 1) In stage 3 of LoCoHD, a subset of primitive atoms is selected as anchor atoms. Could the authors give details of the selection of anchor atoms? Which selection method produces better results?
- 2) How are the weights set in the equation of LoCoHD score for a given anchor atom pair? The authors should provide detailed information.
- 3) In section 2.2 Primitive Typing Schemes, several different primitive typing schemes have been proposed, which one is better? Can the authors give a general solution (as in the case of CASP competitions)?
- 4) LoCoHD has been extensively compared with Iddt, demonstrating the favorable design of LoCoHD. However, what I'm more interested in understanding is how LoCoHD's superiority compares to existing scoring metrics such as IDDT and TM-score. In what scenarios can LoCoHD provide a more reasonable assessment and better model selection?
- 5) The experiments conducted by the authors on structure ensembles are interesting. Currently, methods like AF2 can generate high-quality predicted structures, but these structures often correspond to the same conformational state. When assessing cases where the native structures and predicted structures belong to different conformational states, were the authors able to perform evaluations? Are there any interesting findings from such scenarios that could be shared?

Reviewer #2 (Remarks to the Author):

The authors have introduced the LoCoHD metric, a novel approach designed to compare pairs of residues from two protein structures based on the chemical composition of the atoms in each residue's environment. This concept, which explicitly incorporates chemical atom type information, is distinct and could capture similarities and differences missed by conventional coordinate distance-based metrics such as IDDT and RMSD. If adequately validated, this could be a complementary approach in the comparative analysis of protein structures, in both prediction settings such as CASP, as well as comparing functional effects of residue changes in proteins with the same fold.

However, the authors do not provide adequate evidence for their key claims.

The claim that "The appearance or disappearance of different side-chain interactions (or interaction networks), changes in salt bridges, hydrogen bonds, π -cation interactions, polar-polar contacts, hydrophobic cores" etc. can be captured using count vectors of atom types. An analysis of how the primitive atom types change across such interactions is required.

The claim that "LoCoHD score can distinguish between environments surrounding residues with different physicochemical properties and different environment-organizing behaviors" (line 251). Comparing random pairs of residues from random pairs of proteins followed by pinpointing two examples of low and high scoring pairs does not provide a strong foundation for this claim. I would expect more comprehensive and systematic statistical comparisons of, for example, proteins from the same superfamily with different and similar functions, or residue mutations leading to neutral or significant effects.

While the authors state that LoCoHD results for the Gag-Pol example indicated "the structure is not perturbed by the domain movements - the most significant chemical changes within the domains are observed by three Arg residues occupying different niches as they rotate on the surface of the protein, independent of the large domain fluctuations." (lines 452-455), it would have been good to see some validation of such a claim from other literature. This again could be substantiated using publicly available datasets of verified mutant effects.

Some choices are not explained or justified

It is unclear why only RMSD is shown for the Gag-Pol example, when the presence of a hinge clearly favours the use of IDDT or CAD score.

Why is FA+Cent used for CASP14 comparisons when the model and reference should have the same atoms?

The utility and comprehensiveness of some figures is questionable.

Fig1 cannot be understood without the corresponding text and does not add anything beyond the text.

Fig4 Panel A does not align with its purpose. While the positioning of Lys94 is indicated, the chemical or structural environment around this residue, which seems central to the paper's theme, is not shown. Similarly, having panels D and E with residue names and LoCoHD scores is not useful without the context of where these residues are in the structure and how their environment looks. This diminishes the value of the figure in illustrating your point. As this kind of example is exactly what sets LoCoHD apart from IDDT I would expect many more such examples with clearly labelled figures of the differences in structural environments and each residue-pair labelled with the corresponding LoCoHD/IDDT to see the complementarity between the two. This also applies to the first NMR example where there are areas with high RMSD range, low LoCoHD and vice versa.

Scatter plots with full opacity blue circles are difficult to interpret as it is unclear how many data points are in a certain region.

Insights from figures and tables need to be adequately described in the results. For example, Table 2 has a lot of rows but only 3 are actually referred to in the results.

The text is quite dense and somewhat challenging to follow.

Many sentences are hard to parse and can easily be simplified. e.g lines 232-240, lines 281-283

Details that pertain to the methodology, such as how datasets were sourced and arranged, are found in the Results section instead of in Methods. e.g lines 196-200, lines 275-280, lines 344-362, lines 421-424

Sentences in figure and table legends sometimes belong in Results. e.g. Fig2 legend lines 245-249, Table 2 legend "It can be observed, that the IDDT and LoCoHD scoring systems agree on the order of the five contestants (rows highlighted with yellow). Also, it can be seen that as the quality of the prediction decreases (IDDT decreases, LoCoHD increases) the magnitude of the median SpR value decreases (row highlighted with light blue)."

Apart from supplementary figures 1 and 5, none of the other supplementary figures are referenced in the main text. Same with SText3.

The paper is notably light on citations.

Reviewer #3 (Remarks to the Author):

I co-reviewed this manuscript with one of the reviewers who provided the listed reports as part of the Nature Communications initiative to facilitate training in peer review and appropriate recognition for co-reviewers.

Answers for Reviewer 1:

"The authors developed a metric (LoCoHD) for the highly flexible local composition Hellinger distance, which is based on the chemical composition of local residue environments. Distinguishing between different conformational states remains a challenge, particularly in the protein structure prediction, where it can be difficult to differentiate whether predicted structures are erroneous predictions or simply correspond to different conformational states. Overall, I am quite interested in this work. There are some questions needed to be addressed. See my comments below,"

We would like to thank You for your interest in our work and for your constructive questions and comments regarding the manuscript.

1. In stage 3 of LoCoHD, a subset of primitive atoms is selected as anchor atoms. Could the authors give details of the selection of anchor atoms? Which selection method produces better results?

Anchor atom pairs designate the two locations around which the environmental differences are to be computed. This way, the notion of "better results" is not so straightforward to decipher. For example, the selection, which provides the most information, might be sought. However, this would require the consideration of all possible primitive atom pairs between the two primitive structures, comparing every possible environment in the first structure to every other in the second structure. This results in long runtimes and larger output datasets, which are also harder to analyze. Although this can be useful sometimes, most of the time this approach is excessive and a subjective selection of anchor-pairs suffices, probing the difference of only some high-interest environment-pairs. We would recommend selecting all primitive atoms as anchors and pairing them to their corresponding primitive atom in the other structure in case of same-protein-different-structure comparisons. However, in case of larger proteins this can result in long runtimes, in case of which, the selection of only centroid primitive atoms as anchors may suffice.

These considerations were included to the text at the second paragraph in "Methods: Description of the LoCoHD Algorithm" as follows:

"Since the selection of the anchor atoms defines not only the global spatial resolution and the focus area of the comparison but also its resource-efficiency, the applied anchoring-scheme has to be adapted to the task at hand."

2. How are the weights set in the equation of LoCoHD score for a given anchor atom pair? The authors should provide detailed information.

Throughout this work we used a uniform weight function which was emphasized at the beginning of each subsection in the Result chapter. (Distribution of LoCoHD Scores: lines 192-193, Comparison of CASP14 Contestants: lines 272-273, Comparison of Structure Ensembles: line 349 and

Using LoCoHD for the analysis of a MD simulation: lines 480-481. Line numbers are according to the unrevised manuscript.)

The LoCoHD of all anchor atom pairs were calculated using this setting, although the use of an anchor-type dependent weighting scheme seems a viable next step (this kind of functionality is - as of yet - unavailable in the current version of the LoCoHD package). The present manuscript focuses on the introduction and first evaluation of the metric and suggesting possible fields of applications. Due to the large flexibility of LoCoHD (the choice of primitive typing scheme, anchor atoms, weighting scheme and data aggregation) - which is among its most important merits, in our view - it was not possible to thoroughly exhaust all its different aspects, for example, the thorough examination of the effect of using different weighting schemes. This is, however, among our future plans - but we felt that it was out of scope of this paper.

3. In section 2.2 Primitive Typing Schemes, several different primitive typing schemes have been proposed, which one is better? Can the authors give a general solution (as in the case of CASP competitions)?

While the choice of anchor atoms provides global spatial resolution, and helps focusing on the region of interest, the choice of weighting scheme provides local spatial resolution, allowing the specification of what the user considers to be the environment of an anchor atom, the choice of the primitive typing scheme provides the chemical resolution, a chance to specify which atoms (or atom groups) should be differentiated based on their nature. These are all equally fine-tunable degrees of freedom, for which the optimal setting can strongly depend on the exact structures, use-case and research interest. We proposed four different typing schemes (FA, CG, FA+Cent, CG+Cent). In case of the FA and CG schemes the anchor pairing is only trivial if the two structures to be compared are comprised of the same atoms. In case of the centroid-containing schemes the triviality of anchor pairing is less restricted, needing only for the two structures to have the same number of residues. Also, centroid-containing schemes can be used if residue-sized global resolution suffices, since then only the centroid primitive atoms must be selected as anchors. On the other hand, schemes FA/FA+Cent and CG/CG+Cent produce different number of primitive atoms, providing a way to reduce metric calculation runtimes through the choice of the latter options (resulting in less chemical resolution in exchange). In general, the choice of the correct primitive typing scheme should be made according to practical and target-specific considerations.

We included a short paragraph at the end of Methods: Primitive Typing Schemes to reflect on these points of consideration:

"Selection of the primitive atoms is again, task dependent. The FA and FA+Cent schemes provide the most chemical resolution. In the case of FA and CG, anchor pairing is only trivial if the two structures to be compared are comprised of the same atoms. Centroid-containing schemes (like FA+Cent and CG+Cent) can be used if residue-sized global spatial resolution suffices, and they also offer a way to reduce the runtime of the metric calculation."

And a sentence at the end of the first paragraph of the Discussion:

"Specification of the task is completed in three conceptually separate steps: the choice of anchor atoms defines global spatial resolution and helps focusing on the region of interest; the choice of weighting scheme sets the local spatial resolution, allowing the specification of what the user considers to be the environment of an anchor atom; and the choice of the primitive typing scheme provides the chemical resolution, a chance to specify which atoms (or atom groups) should be differentiated based on their nature."

4. LoCoHD has been extensively compared with lddt, demonstrating the favorable design of LoCoHD. However, what I'm more interested in understanding is how LoCoHD's superiority compares to existing scoring metrics such as lDDT and TM-score. In what scenarios can LoCoHD provide a more reasonable assessment and better model selection?

We do not think of LoCoHD as a superior technique as compared to lDDT or TM-score, but rather as a technique providing orthogonal information about the structure pair to be compared. While RMSD, lDDT, TM-score, GDT-TS, and other scoring systems focus on differences derived from geometrical considerations, LoCoHD focuses on differences derived from the local chemical composition. Model selection based on the classical tools supplemented with LoCoHD can provide a more refined picture about model quality. As an example: a residue in a predicted model (compared to a reference structure) can have simultaneously low lDDT (showing low model quality) but also low LoCoHD (showing high model quality). This would mean that although the two structures show high geometrical differences, the chemical composition around the residue remains the same. Thus, the predictor producing the model may not assign the correct interacting partners to the residue in question, but "understands" the chemical nature that should surround it. We would argue that this metric can help us understand how we can further improve a predictor's performance in terms of its internal physico-chemical knowledge. Also, LoCoHD can be used to estimate the functional severity of structural fluctuations - conformational changes that cause only minor variation in the LoCoHD of the active site or other regions of interest are less expected to be of great significance, irrespective of their spatial extent. As in case of the example of Gag-Pol (in Results: Comparison of Structure Ensembles Through LoCoHD and RMSD), we were able to conclude that the extensive domain movements do not create chemical differences within the domains themselves.

5. The experiments conducted by the authors on structure ensembles are interesting. Currently, methods like AF2 can generate high-quality predicted structures, but these structures often correspond to the same conformational state. When assessing cases where the native structures and predicted structures belong to different conformational states, were the authors able to perform evaluations? Are there any interesting findings from such scenarios that could be shared?

Although not in connection to the AF2 predictions, but a similar case-study was present in the manuscript, namely the structure-clusterings in the

LoCoHD-RMSD comparison. RMSD clustering of structures is a popular method to separate different conformers of the same protein, for example in molecular dynamics trajectory analysis or in case of NMR ensembles. We performed structural clustering based on both RMSD and LoCoHD and, indeed, found interesting results. The results indicate that in case of small structural changes, RMSD and LoCoHD clusters the slightly different conformers in the same manner. However, in case of large conformational changes, RMSD and LoCoHD identify different clusters. This is to be expected, since RMSD is a more stringent metric, demanding global correctness. In contrast, LoCoHD focuses on local changes, which only have to be correct on a "chemical" level and can be incorrect on the "exact atomic id" level. We think that this "permissiveness" of the new metric causes it to only differentiate conformers significantly if the interaction-types of their residues is also highly different. Different clustering results also mean that among the conformers which appear to be similar based on their spatial coordinates, LoCoHD can select those that may be worth of individual consideration. For example, using LoCoHD we were able to identify the previously undetected significance of the His276 residue in the overall conformational freedom of podocin (which we show in Results: Using LoCoHD for the analysis of an MD Simulation). Podocin is member of the kidney's filtration system, and the severity of its pathological mutations was linked to the flexibility - or pliability - of its monomeric units when forming its functional, multimeric form.

We added a sentence referring to the possible functional significance of this finding to the aforementioned section:

"Since the 273-277 segment is in the critical hinge region of the podocin monomer, the flexibility of which was suggested to influence the effect that pathological mutations exert (<https://pubmed.ncbi.nlm.nih.gov/24509478/>), recognizing that two interaction-wise different orientations are sampled by His276 may carry functional significance."

In addition, we found evidence for the importance of Arg150 in case of the HIV-1 Gag-Pol example (mentioned in section Results: Comparison of Structure Ensembles Through LoCoHD and RMSD), where the conformational changes of this residue can be attributed to the changes in the composition in its environment. This arginine plays a crucial role in the capsid formation of HIV-1, as pointed out by several sources (<https://pubmed.ncbi.nlm.nih.gov/17923088/>, <https://pubmed.ncbi.nlm.nih.gov/27129282/>, <https://pubmed.ncbi.nlm.nih.gov/30069050/>). We inserted a small text into our manuscript regarding this topic, which can be found at the end of the referred section:

"Interestingly, one of these, Arg150 (or Arg18, according to a different numbering convention), was shown crucial for the formation of the hexameric capsid of HIV-1. Mutations at this site result in distinct morphological variation of the viral assembly without causing conformational changes discernible by solid state NMR. LoCoHD identified this residue as being able to detect conformational fluctuations of the matrix – as would be

expected of a residue that recognizes the presence of interaction partners and guides the assembly process." _____

Answers for Reviewer 2:

"The authors have introduced the LoCoHD metric, a novel approach designed to compare pairs of residues from two protein structures based on the chemical composition of the atoms in each residue's environment. This concept, which explicitly incorporates chemical atom type information, is distinct and could capture similarities and differences missed by conventional coordinate distance-based metrics such as IDDT and RMSD. If adequately validated, this could be a complementary approach in the comparative analysis of protein structures, in both prediction settings such as CASP, as well as comparing functional effects of residue changes in proteins with the same fold."

Thank You for the thorough consideration of our manuscript and the valuable suggestions.

Method validation is incomplete and several claims are not validated:

1. The claim that "The appearance or disappearance of different side-chain interactions (or interaction networks), changes in salt bridges, hydrogen bonds, π -cation interactions, polar-polar contacts, hydrophobic cores" etc. can be captured using count vectors of atom types. An analysis of how the primitive atom types change across such interactions is required.

Thank You for your insight on the missing validation. Indeed, this statement was poorly supported, which is why we added a new paragraph to section Results: Distribution of LoCoHD scores. We supplemented the previous analyses with residue interaction calculations and connected these results with the LoCoHD score through a neural network model. It was possible to show that there is a connection between the count of the different types of interactions belonging to the central residues within local environments and the LoCoHD score of these environments. Using only the central residue type information and the counts of the different residue-residue interactions, the network was able to learn to predict the LoCoHD score with a mean absolute error of 4.5 % and a SpR of 0.452, proving the dependence between its inputs and outputs. This result also shows that the LoCoHD score contains more information that could be gained by simply counting the residue-residue contacts of each amino acid (in which case the training would lead to perfect prediction of the LoCoHD score). These new results are presented in detail in the revised version of the manuscript.

2. The claim that "LoCoHD score can distinguish between environments surrounding residues with different physicochemical properties and different environment-organizing behaviors" (line 251). Comparing random pairs of residues from random pairs of proteins followed by pinpointing two examples of low and high scoring pairs does not provide a strong foundation for this claim. I would expect more comprehensive and systematic statistical comparisons of, for example, proteins from the same superfamily with different and similar functions, or residue mutations leading to neutral or significant effects.

This claim is - at least in part - supported by the previous answer (and the analysis we included in the text), but we also carried out comparisons of different serine proteases, where single mutations have significant or neutral effects - as You suggested. These are available in **Supplementary Note 3-4** and in **Supplementary Figure 6-9**.

3. While the authors state that LoCoHD results for the Gag-Pol example indicated "the structure is not perturbed by the domain movements - the most significant chemical changes within the domains are observed by three Arg residues occupying different niches as they rotate on the surface of the protein, independent of the large domain fluctuations." (lines 452-455), it would have been good to see some validation of such a claim from other literature. This again could be substantiated using publicly available datasets of verified mutant effects.

Thank You for pointing this out. The critical significance of the Arg150 residue in formation of the hexamer capsid of HIV-1 virus was previously demonstrated (<https://pubmed.ncbi.nlm.nih.gov/17923088/>, <https://pubmed.ncbi.nlm.nih.gov/27129282/>, <https://pubmed.ncbi.nlm.nih.gov/30069050/>) (using a different numbering, these manuscripts refer to Arg150 as Arg18). Mutations at this site cause distinct morphological changes in the assembly, but "with only small differences in (15)N and (13)C chemical shifts, no significant differences in NMR line widths, and few differences in the number of detectable NMR cross-peaks. Thus, the pronounced differences in morphology do not involve major differences in the conformations and identities of structurally ordered protein segments." (<https://pubmed.ncbi.nlm.nih.gov/27129282/>) So, it is really exciting to see, that LoCoHD identifies this residue as being able to detect conformational fluctuations of the matrix - as would be expected of a residue that recognizes the presence of interaction partners and guides the assembly process. Arg299 (or Arg167) on the other hand was first suspected to be a member of an H-bond network that is vital to the stability of the fold - based on the crystal structure of the C-terminal domain of Gag (<https://pubmed.ncbi.nlm.nih.gov/9346481/>) - but was later shown to fluctuate between the H-bonded and solvated conformations based on solution-state NMR results (<https://pubmed.ncbi.nlm.nih.gov/18417468/>). This is also in line with our findings.

We included references to these experimental results in the revised version of the manuscript along with a short summary:

"Interestingly, one of these, Arg150 (or Arg18, according to a different numbering convention), was shown crucial for the formation of the hexameric capsid of HIV-1. Mutations at this site result in distinct morphological variation of the viral assembly without causing conformational changes discernible by solid state NMR. LoCoHD identified this residue as being able to detect conformational fluctuations of the matrix – as would be expected of a residue that recognizes the presence of interaction partners and guides the assembly process."

Some choices are not explained or justified.

4. It is unclear why only RMSD is shown for the Gag-Pol example, when the presence of a hinge clearly favors the use of IDDT or CAD score.

In this article LoCoHD is extensively compared to other metrics, like RMSD, IDDT, and now - due to your mention and suggestion - to the CAD score too (see **Supplementary Table 2-3**). In the ensemble section (section "Comparison of Structure Ensembles Through LoCoHD and RMSD") we wanted to focus on the possible connections between LoCoHD and RMSD. Indeed, in the Gag-Pol example the presence of the hinge region would make the use of scores IDDT or CAD more justified compared to that of the RMSD score, but most of the time the quality analysis of NMR ensembles is still done using RMSD. Due to its popularity, we wanted to assess the connection of LoCoHD with RMSD through several examples: in case of the E5 miniprotein, where the use of RMSD is justified, in case of the Gag-Pol two-domain protein, where local measures, like IDDT, CAD, or LoCoHD give more informative results, and in case of the IDP cGMP phosphodiesterase, for which both local and global measures are hard to decipher. We wanted to give the reader a complete picture by the inclusion of all of these test cases.

5. Why is FA+Cent used for CASP14 comparisons when the model and reference should have the same atoms?

This example was also used to demonstrate the connection between the behaviour of LoCoHD and the behaviour of IDDT. Since IDDT is given as a per-residue metric, we also wanted to use LoCoHD in a residue-wise and not in an atom-wise fashion.

[Problems with figures and tables]

6. Fig1 cannot be understood without the corresponding text and does not add anything beyond the text.

We feel that Figure 1 might be useful for some readers as a simple overview of the LoCoHD process, but we also included an extended version in the Supplementary document (**Supplementary Figure 1**) that contains more information.

7. Fig4 Panel A does not align with its purpose. While the positioning of Lys94 is indicated, the chemical or structural environment around this residue, which seems central to the paper's theme, is not shown. Similarly, having panels D and E with residue names and LoCoHD scores is not useful without the context of where these residues are in the structure and how their environment looks. This diminishes the value of the figure in illustrating your point. As this kind of example is exactly what sets LoCoHD apart from IDDT I would expect many more such examples with clearly labelled figures of the differences in structural environments and each residue-pair labelled with the corresponding LoCoHD/IDDT to see the complementarity between the two. This also applies to the first NMR example where there are areas with high RMSD range, low LoCoHD and vice versa.

Indeed, the figure could have been better prepared. It is now modified in the revised article, hopefully providing a clearer view.

8. Scatter plots with full opacity blue circles are difficult to interpret as it is unclear how many data points are in a certain region.

Thank You for your note! The opacity of the scatter plots were adjusted for easier visualization.

9. Insights from figures and tables need to be adequately described in the results. For example, Table 2 has a lot of rows but only 3 are actually referred to in the results.

Thank You for your suggestion! Rows that were not discussed in the main text were moved to the **Supplementary Information**.

[Other notes]

10. The text is quite dense and somewhat challenging to follow. Many sentences are hard to parse and can easily be simplified. e.g lines 232-240, lines 281-283.

We attempted to disambiguate the text wherever possible.

11. Details that pertain to the methodology, such as how datasets were sourced and arranged, are found in the Results section instead of in Methods. e.g lines 196-200, lines 275-280, lines 344-362, lines 421-424.

Thank You for your insight! We moved the mentioned lines to the Materials and Methods section, except for lines 421-424, since these explain protein-specific information and not general methodology.

12. Sentences in figure and table legends sometimes belong in Results. e.g. Fig2 legend lines 245-249, Table 2 legend "It can be observed, that the lDDT and LoCoHD scoring systems agree on the order of the five contestants (rows highlighted with yellow). Also, it can be seen that as the quality of the prediction decreases (lDDT decreases, LoCoHD increases) the magnitude of the median SpR value decreases (row highlighted with light blue)."

Regarding the quoted text from Table2, this explanation is also present in Results: the text "It can be observed, that the lDDT and LoCoHD scoring systems agree on the order of the five contestants [...]" is mentioned in Results as "Our first results show an agreement between the contestant-order set by the median lDDT and LoCoHD values, [...]", while the text "it can be seen that as the quality of the prediction decreases (lDDT decreases, LoCoHD increases) the magnitude of the median SpR value decreases [...]" corresponds to the part "These tendencies inherently create correlations between different scoring systems, with higher absolute correlations closer to similarity extremities." in Results. For the referenced text in Figure 2 we provided an additional paragraph in section Results: Distribution of LoCoHD scores. We believe that a figure along with its legend should behave like a

singleton and should be easy to parse without the main text, hence the inclusion of result-elements.

13. Apart from supplementary figures 1 and 5, none of the other supplementary figures are referenced in the main text. Same with SText3.

Thank You for your remark, the missing supplementary references were placed in the main text.

14. The paper is notably light on citations.

Thank You for pointing this out, new and relevant references were added to the manuscript.

Reviewer #1 (Remarks to the Author):

I support publication, but before that, I have a few minor questions that I would appreciate the author answering.

The author compared LoCoHD and IDDT on the results of CASP14, so why not consider the latest CASP15? I am very interested in the performance of LoCoHD on complexes, especially antibody-antigen(H1166-H1168) and nanobody-antigen(H1140-H1144). If possible, I hope the author can use the CASP15 target as an example to conduct an analysis.

Some sentences in the manuscript and the descriptions of figures and tables also seem not to be very easy to understand, and it is hoped that this can be improved.

The code provided does not seem to execute well, please check further.

Reviewer #1 (Remarks on code availability):

The code provided does not seem to execute well, please check further.

Reviewer #2 (Remarks to the Author):

Overall the authors addressed most of my comments. In particular, the analysis of significant and neutral mutations in the Supplementary showcases and highlights the merits of LoCoHD over distance-based scores. Some remaining suggestions:

* While the figures are now improved, I still believe that some figures could benefit from the addition of specific interaction types, especially in examples where these interactions are explicitly mentioned (e.g lines 435, 598 in main text and supplementary note 4). The authors could take a look at tools such as PLIP which visualises such interactions in PyMol.

* Lines 343-345: "Some residues are arranged in "arm"-like patterns, where there is one amino acid close to the center of the plot (e.g. Ser, Thr, Val), followed by 1-4 others, gradually getting farther away. These "arms" include the series Glu-Gln-Lys-Arg, Ile-Leu-Phe-Tyr-Trp, or Met-His." The spatial patterns in non-linear embeddings such as t-SNE cannot be interpreted in this fashion, as shown for example in <https://journals.plos.org/ploscompbiol/article?id=10.1371/journal.pcbi.1011288>. Instead would rephrase as these residue types cluster close together.

*I would recommend replacing Figure 1 with Supplementary Figure 1 as it provides more detail into the approach and is more standalone.

Minor suggestions:

* Lines 304-310 could be rephrased as: "Statistical descriptors for the different residue type pairs were also extracted from the random samples. The residue type pairs with the highest and lowest average LoCoHD scores are shown in Table 1 for the primitive typing scheme FA+Cent. A t-distributed stochastic neighborhood embedding (tSNE) was also performed using the mean LoCoHD of hetero-residue pairs (i.e. where the residue types are not the same)."

* Standard deviation values could be added back to Table 2 in the same row as the median using the \pm notation (i.e median \pm stdev)

* Lines 623-626 could be rephrased as: "The 273-277 segment is in the critical hinge region of the podocin monomer, the flexibility of which was suggested to influence the effect that pathological mutations exert⁵². Thus, recognizing that two interaction-wise different orientations are sampled by His276 may carry functional significance."

* In Supplementary Note 1: DeeplyTough is misspelt as DeeplyThought

Answers for Reviewer #1:

I support publication, but before that, I have a few minor questions that I would appreciate the author answering.

1. The author compared LoCoHD and IDDT on the results of CASP14, so why not consider the latest CASP15? I am very interested in the performance of LoCoHD on complexes, especially antibody-antigen (H1166-H1168) and nanobody-antigen (H1140-H1144). If possible, I hope the author can use the CASP15 target as an example to conduct an analysis.

Thank you for your suggestion! We performed a similar analysis on the CASP15 database as was carried out on the CASP14 database. The results for these are collected in **Supplementary Table 4** and in **Supplementary Figures 11-15**. Also, in light of these new findings, we added a new paragraph at the end of the "Results/Comparison of CASP14 Contestants Through LoCoHD and IDDT" sub-chapter.

From a global perspective, competitors TS229, TS278, TS439 perform very closely, followed by TS074. This can be seen both by the IDDT and the LoCoHD analysis, which suggest the same contestant-order. Since these predictors perform similarly well in CASP15 as AlphaFold2 did in CASP14, the IDDT values are in the high regime, causing a high IDDT-LoCoHD correlation (a median SpR between -0.68 and -0.72) (see main text, same sub-chapter, paragraph 3).

At the level of individual residues, the usefulness of LoCoHD is more pronounced. As per your suggestion, we focused on H1166 and H1144 for a more thorough analysis and found interesting results. For both target structures we analyzed TS278's predicted structures. In the case of H1166, we showcased the data concerning two residues; Arg109 from chain B (the antibody) and Tyr80 from chain C (the antigen). In H1144, we looked at the environments of residue Arg105 from chain B (the nanobody). In all of these cases, the large LoCoHD change can be attributed to changes in the aromatic content of the local environments, showing the high sensitivity of the FA+Cent typing scheme to these types of interactions. This is not so surprising, since with all heavy atoms included as primitive atoms, a single aromatic residue can contribute with a large primitive type count to the environment, which consequently affects the DDEC function values more than in the case of other functional groups, creating a slight bias toward aromatic interactions, which – on the other hand – has a sound biophysical basis (see, for example, <https://www.nature.com/articles/s41586-022-04417-6> and <https://jcheminf.biomedcentral.com/articles/10.1186/s13321-020-00437-4>). Nevertheless, visual inspection and IDDT analysis of the H1144 experimental and predicted structures reveals the correctly docked nanobody has the lower median prm-LoCoHD score.

2. Some sentences in the manuscript and the descriptions of figures and tables also seem not to be very easy to understand, and it is hoped that this can be improved.

We thoroughly reviewed the text and attempted to express definitions and explanations as clear as possible. Certain hard-to-parse sentences were also rephrased.

3. The code provided does not seem to execute well, please check further.

Thank you for the feedback! We extended the execution options with a containerized version of LoCoHD using Docker and also with an option to run LoCoHD as a CLI tool. Please check the README.md file in the GitHub repository at https://github.com/fazekasz/loco_hd for further information. Note, that in order to compile LoCoHD, the maturin package is needed, which only works inside a Python virtual environment (this has also been indicated now in the README file) and if a Rust compiler is present. Also, to run the Python scripts that reproduce our results, you also need to download the datasets used by these scripts (from Figshare (see the **Data availability** section) or from the original sources, like the CASP archive). It is also necessary to set the corresponding script variables pointing to these files. Note, that while LoCoHD itself is intended to work as a general tool, the example scripts are only intended to work with the provided datasets, and are not necessarily able to generalize to other datasets. For fast development and bugfixes, it is also possible to file an issue under the Issues tab on GitHub, specifying the exact problem.

Answers for Reviewer #2:

Overall the authors addressed most of my comments. In particular, the analysis of significant and neutral mutations in the Supplementary showcases and highlights the merits of LoCoHD over distance-based scores. Some remaining suggestions:

1. While the figures are now improved, I still believe that some figures could benefit from the addition of specific interaction types, especially in examples where these interactions are explicitly mentioned (e.g lines 435, 598 in main text and supplementary note 4). The authors could take a look at tools such as PLIP which visualises such interactions in PyMol.

Thank you for the suggestion! Indeed, we found PLIP to be an excellent tool for the visualization of residue-residue interactions. Several structure depicting figures were updated with PLIP visualizations.

2. Lines 343-345: "Some residues are arranged in "arm"-like patterns, where there is one amino acid close to the center of the plot (e.g. Ser, Thr, Val), followed by 1-4 others, gradually getting farther away. These "arms" include the series Glu-Gln-Lys-Arg, Ile-Leu-Phe-Tyr-Trp, or Met-His." The spatial patterns in non-linear embeddings such as t-SNE cannot be interpreted in this fashion, as shown for example in <https://journals.plos.org/ploscompbiol/article?id=10.1371/journal.pcbi.1011288>. Instead would rephrase as these residue types cluster close together.

Thank you, this was an eye-opening article about the drawbacks of dimension-reduction methods. The aforementioned part was rephrased.

3. I would recommend replacing Figure 1 with Supplementary Figure 1 as it provides more detail into the approach and is more standalone.

Although we concur that **Supplementary Figure 1** is more expressive, we would argue that the size of this figure does not conform well with the formatting guidelines set by Nature Communications, and would keep it in the supplementary document.

Minor suggestions:

4. Lines 304-310 could be rephrased as: "Statistical descriptors for the different residue type pairs were also extracted from the random samples. The residue type pairs with the highest and lowest average LoCoHD scores are shown in Table 1 for the primitive typing scheme FA+Cent. A t-distributed stochastic neighborhood embedding (tSNE) was also performed using the mean LoCoHD of hetero-residue pairs (i.e. where the residue types are not the same)."

Thank you for your suggestion. The rephrased version was incorporated into the main text.

5. Standard deviation values could be added back to Table 2 in the same row as the median using the \pm notation (i.e median \pm stdev)

Thank you! **Table 2** was populated with the corresponding StdDev values.

6. Lines 623-626 could be rephrased as: "The 273-277 segment is in the critical hinge region of the podocin monomer, the flexibility of which was suggested to influence the effect that pathological mutations exert⁵². Thus, recognizing that two interaction-wise different orientations are sampled by His276 may carry functional significance."

Thank you for your suggestion. The rephrased version was incorporated into the main text.

7. In Supplementary Note 1: DeeplyTough is misspelt as DeeplyThought

Thank you for the note! The typo was corrected.

Reviewer #1 (Remarks to the Author):

I support publication.

Reviewer #2 (Remarks to the Author):

I assume the abstract refers to the "CASP" - not "CAPS" website: "... IDDT scores provided by the CAPS website".

Otherwise I have no further comments.

Reviewer #2 (Remarks on code availability):

For the second revision, only changes in the manuscript were reviewed.

Answers to the Reviewers

Reviewer #1:

I support publication.

We would like to thank you for your kind support and constructive comments throughout the whole revision process.

Reviewer #2:

I assume the abstract refers to the "CASP" - not "CAPS" website: "... IDDT scores provided by the CAPS website".

Otherwise I have no further comments.

Thank you for the correction, the mentioned part has been corrected. We would also like to thank you for your kind support and constructive comments throughout the whole revision process.